# Discrete Diffusion VLA: Bringing Discrete Diffusion to Action Decoding in Vision-Language-Action Policies

**Zhixuan Liang**[1,2]  **Yizhuo Li**[1]  **Tianshuo Yang**[1,2]  **Chengyue Wu**[1]  **Sitong Mao**[4]  **Liuao Pei**[1]
**Tian Nian**[2]  **Shunbo Zhou**[5]  **Xiaokang Yang**[3]  **Jiangmiao Pang**[2]  **Yao Mu**[3,2]  **Ping Luo**[1]

## Abstract

Vision–Language–Action (VLA) models adapt large vision–language backbones to map images and instructions into robot actions. However, prevailing VLAs either generate actions autoregressively in a fixed left-to-right order with poor performance or attach separate diffusion heads outside the backbone that fragments information pathways and hinders unified, scalable architectures. Instead, we present Discrete Diffusion VLA that discretizes action chunks and models them with discrete diffusion pattern retaining progressive refinement inside the unified transformer backbone. Our method achieves an adaptive decoding order that resolves high-confidence action elements before harder ones and employs secondary re-masking to revisit uncertain predictions, enabling robust error correction. This design preserves pretrained vision-language priors, supports parallel decoding, and improves the efficiency. Discrete Diffusion VLA achieves 96.4% avg. success on LIBERO, 71.2% visual matching on SimplerEnv-Fractal, and 54.2% overall on SimplerEnv-Bridge. On out-of-distribution tests of LIBERO-Goal, our method exhibits only 0.8% language degradation versus 8.0% of parallel decoding, and 20.4% vision degradation versus 29.0% for continuous diffusion, demonstrating well retention of pretrained vision-language capabilities. We also conduct two real-robot evaluations on AgileX Cobot Magic platform to show the method's effectiveness.

---

[1]The University of Hong Kong [2]Shanghai AI Laboratory [3]Shanghai Jiao Tong University [4]Huawei Cloud Computing Technologies Co., Ltd. [5]Ola Dimensions. Correspondence to: Ping Luo <pluo@cs.hku.hk>, Yao Mu <muyao@sjtu.edu.cn>.

*Proceedings of the $43^{rd}$ International Conference on Machine Learning*, Seoul, South Korea. PMLR 306, 2026. Copyright 2026 by the author(s).

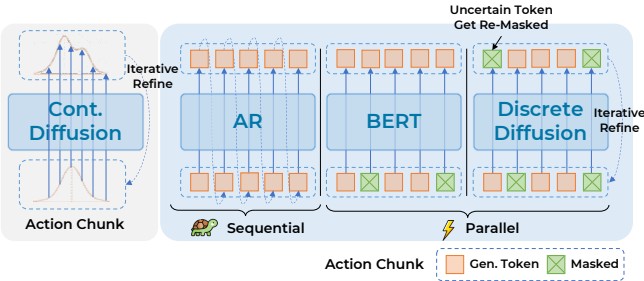

*Figure 1.* **Paradigm comparison.** Continuous diffusion over action chunks (left) versus discrete token decoders: AR (sequential), BERT-style (parallel), and our discrete diffusion with re-masking.

## 1. Introduction

Vision-Language-Action (VLA) models enable robots to interpret visual and linguistic inputs and execute corresponding action sequences. Modern VLA frameworks typically adapt a large pretrained vision-language model (VLM) by adding an action-generation head that outputs motor commands (either continuous trajectories or discrete tokens). Current approaches fall into two paradigms: (1) an autoregressive (AR) approach, inspired by GPT-style transformers, that predicts discretized action tokens sequentially (*e.g.* OpenVLA (Kim et al., 2024), $\pi_0$-FAST (Pertsch et al., 2025)); and (2) a separate action head that employs MLP or continuous diffusion to map VLM output latent tokens to executable actions (*e.g.*, $\pi_0$ (Black et al., 2024) and SmolVLA (Shukor et al., 2025)). Continuous diffusion paradigm is more dominant as it can better model sophisticated actions than AR, but is decoupled from the VLM backbone. Although some recent works integrate it into one architecture (*e.g.*, Transfusion (Zhou et al., 2024) in $\pi_0$), they still rely on diffusion-specific training and optimization goal, having competing gradient signals with the VLM component. This design not only complicates policy training but also degrades the pretrained vision-language capabilities, which represents a critical issue we address in this work.

Drawing on recent advances in discrete diffusion and discrete flow-matching for language and multi-modal generation (Nie et al., 2025a; Shi et al., 2024b; Gat et al., 2024; Yang et al., 2026a), we introduce Discrete Diffusion VLA,

a VLA framework that employs discrete diffusion for action generation that unifies vision, language, and action modeling within a single transformer and using consistent optimization goal. This VLA policy is designed to achieve high action precision while preserving strong VLM priors.

In our method, each action dimension is first discretized into tokens via a binning scheme and grouped into chunks with fixed length which is naturally suited for discrete diffusion's block-wise parallel decoding manner. During training, a subset of tokens are masked in the action chunk first and the policy is trained to predict them from the context of unmasked ones across all modalities. At inference, the method starts with all action tokens masked, concatenated with visual and language inputs, and iteratively predicts and re-masks low-confidence tokens until convergence, following a philosophy of "walk before you run". We further employ a secondary re-masking mechanism to enforce consistency across denoising steps, yielding flexible parallel decoding and robust error correction.

A key advantage of Discrete Diffusion VLA over continuous diffusion heads is its superior retention of pretrained vision-language capabilities. Our work achieves it by performing action generation with consistent training objective inside the VLM backbone, exhibiting only 0.8% language degradation versus diffusion's 2.4%, and 20.4% vision degradation versus diffusion's 29.0% on OOD tests of LIBERO-Goal, consistent with prior findings that diffusion-based VLAs are often overly vision-dependent. On the other hand, AR models also unify action generation in VLM backbone, but they suffer from left-to-right compounding error, inference inefficiency and inability to utilize information from later action tokens of the same chunk. Our method instead decode action chunks in parallel over a fixed number of steps, eliminating linearly scaled forward passes and leveraging full modality context for action prediction.

We evaluate Discrete Diffusion VLA on a Franka Panda arm (LIBERO (Liu et al., 2023)), a Google Robot (SimplerEnv-Fractal (Li et al., 2025)), and a WidowX Arm (SimplerEnv-Bridge (Li et al., 2025)), using only RGB inputs, language instructions, and end-effector positions (no depth or affordances). On LIBERO, we achieve 96.4% average success (best among discretized action methods, 0.7% behind the overall continuous SOTA considering the inherent loss from bin-based tokenization, while exhibiting superior OOD robustness). On SimplerEnv, Discrete Diffusion VLA achieves SOTA across both discrete and continuous methods that 71.2% visual matching with 64.1% overall on Fractal, and 54.2% overall on Bridge (+14.7% over $\pi_0$, +6.4% over $\pi_0$-FAST). Besides, we also conduct real-robot evaluations on Cobot Magic. Discrete Diffusion VLA outperforms the other baselines on two tabletop manipulation tasks at 9.69 Hz. Visualizations confirm that the learned de-

coding order adaptively prioritizes high-confidence tokens, revealing interpretable refinement patterns.

In summary, our contributions are threefold: 1) We introduce the first discrete diffusion VLA, unifying action generation with vision-language modeling in one transformer, demonstrating superior retention of pretrained VL capabilities. 2) We develop an adaptive decoding strategy with secondary re-masking that enables confidence-based action-token decoding and robust error correction, improving both effectiveness and efficiency. 3) We validate Discrete Diffusion VLA on three simulation benchmarks and a real robot platform, achieving SOTA on SimplerEnv (Fractal 64.1%, Bridge 54.2%) across all methods, and 96.4% on LIBERO with superior OOD robustness of vision-language retention over continuous SOTA. Comprehensive ablations and visualizations confirm our design.

## 2. Related Works

### 2.1. Vision-Language-Action Models

Early VLA systems (*e.g.*, RT-1 and RT-2 (Brohan et al., 2023; Zitkovich et al., 2023)) adopted a two-part form design that used VLM to produce latent tokens, with a separate MLP decoder mapping them to discretized controls. Subsequent work shifted to AR architecture with scaled backbones (Beyer et al., 2024; Bai et al., 2025; Fang et al., 2026b) for general manipulation (*e.g.* OpenVLA) (Kim et al., 2024; Touvron et al., 2023b; Oquab et al., 2024; Zhai et al., 2023). For high-frequency control, token compression and action chunking further improve policy effectiveness (*e.g.* $\pi_0$-FAST (Pertsch et al., 2025; Kim et al., 2025b)). In parallel, diffusion and flow-matching action heads model continuous trajectories (Janner et al., 2022; Chi et al., 2023; Liu et al., 2025; Li et al., 2024a; Liang et al., 2023), with hierarchical designs (*e.g.* $\pi_0/\pi_{0.5}$) (Black et al., 2024; Intelligence et al., 2025; Bu et al., 2025b;a; Liang et al., 2024; Zhong et al., 2026; Liu et al., 2026; Fang et al., 2026a) and gradually become the dominant pattern in VLA that employ a separate denoising head conditioned on VL tokens.

In this work, by contrast, we perform discrete diffusion over tokenized action chunks inside the VLM backbone with consistent optimization gradients, enabling parallel, revisable decoding via iterative re-masking, and preserving the VLM's emergent multimodal capabilities.

### 2.2. Discrete Diffusion Models

Discrete diffusion has achieved strong results on tokenized images and natural language. Foundational work like D3PM (Austin et al., 2021) formalizes discrete diffusion as a Markov chain that factorizes across positions, where each token is independently corrupted into a categorical distribution. VQ-Diffusion (Gu et al., 2022) and

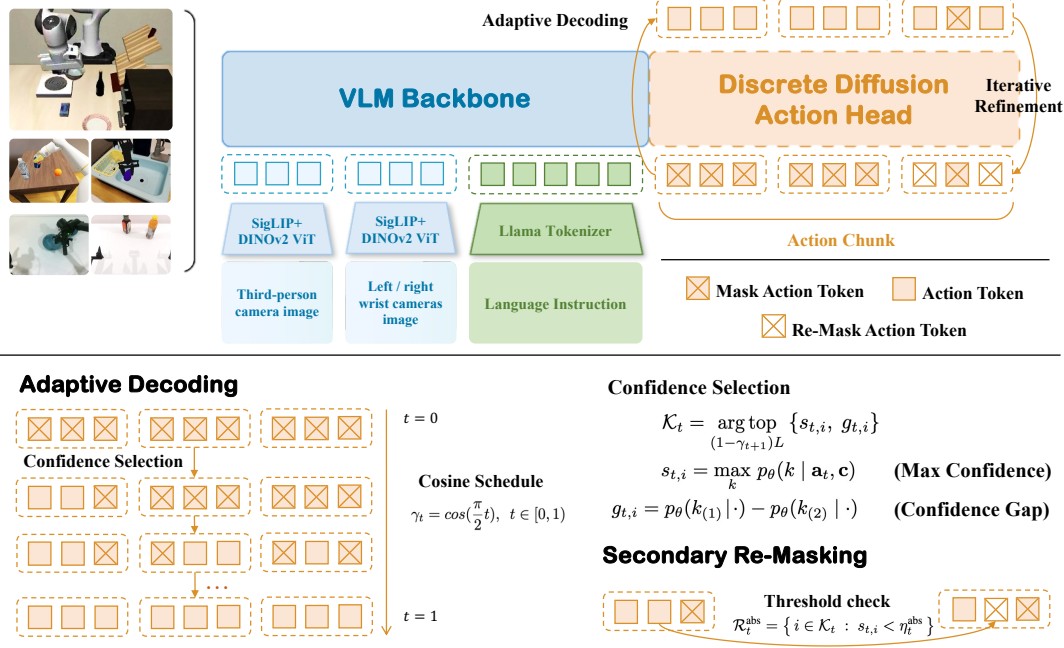

*Figure 2.* **Overview of Discrete Diffusion VLA architecture.** We extend the VLM backbone that encodes multi-view RGB images (SigLIP+DINOv2 ViTs) and linguistic instruction to decode discrete action chunks via diffusion-style iterative refinement. *Adaptive Decoding* (bottom left) keeps high-confidence tokens each round and anneals the keep ratio with a cosine schedule. *Secondary Re-Masking* (bottom right) uses threshold check to re-mask uncertain tokens, enforcing multi-iteration consistency and robust error correction.

MaskGIT (Chang et al., 2022a) build on this for high-fidelity image generation with transformers that iteratively predict masked/corrupted image tokens. In language, Diffusion-BERT (He et al., 2023) and Masked Diffusion Models (Shi et al., 2024c; Zheng et al., 2025a) demonstrate viability, while LLaDA (Nie et al., 2025b) and DiffuLLaMA (Gong et al., 2025) scale to 7B parameters competitive with AR baselines. MMaDA (Yang et al., 2026a) further shows unified discrete diffusion can jointly generate text and images.

Our work extends this line to the *action* modality. We perform discrete diffusion on chunked action tokens, yielding competitive VLA performance while preserving vision-language capabilities.

## 3. Discrete Diffusion VLA Model

### 3.1. Overview

Discrete Diffusion VLA is illustrated in Figure 2. Given image observations (single- or multi-view) and a language instruction, the model extends a VLM backbone to generate actions via discrete diffusion. A unified transformer jointly attends to visual features, language embeddings, and partially unmasked action tokens, progressively demasking remaining masked action tokens according to a diffusion denoising schedule. This formulation eliminates the competing gradients introduced by a separate diffusion loss, preserving VLM priors while unifying perception, instruction

grounding, and action denoising within a single backbone. Section 3.2 formulates discrete diffusion over actions. Section 3.3 describes the model architecture. Section 3.4 details the training pipeline. Section 3.5 covers inference with adaptive decoding and secondary re-masking.

### 3.2. Formulation of Discrete Diffusion over Actions

Let a single action chunk be a length-$L$ token sequence $\mathbf{a}_0 = (a_{0,1}, \ldots, a_{0,L})$, where each $a_{0,i} \in \{1, \ldots, K\}$ is a discretized action token over a vocabulary of $K$ bins. We augment this vocabulary with a special mask token [MASK], yielding $V = K{+}1$ symbols with one-hot basis $\{\mathbf{e}_1, \ldots, \mathbf{e}_K, \mathbf{e}_M\}$, where $\mathbf{e}_k \in \mathbb{R}^V$ has 1 at position $k$ and 0 elsewhere.

The **forward** (noising) process of discrete diffusion is a Markov chain $\{\mathbf{a}_t\}_{t=0}^T$ with per-step transition matrices $\mathbf{Q}_t \in \mathbb{R}^{V \times V}$ that independently map each token to M with probability $\beta_t$ and keep it unchanged with probability $1{-}\beta_t$. Formally, for any one-hot vector $\mathbf{e}_{a_{t,i}}$ of token $a_{t,i}$,

$$a_{t+1,i} \sim \text{Categorical}(\mathbf{Q}_t \, \mathbf{e}_{a_{t,i}}),$$
$$\mathbf{Q}_t \, \mathbf{e}_{a_{t,i}} = (1{-}\beta_t) \, \mathbf{e}_{a_{t,i}} + \beta_t \, \mathbf{e}_M. \tag{1}$$

Composing transition matrices yields $\bar{\mathbf{Q}}_t = \mathbf{Q}_t \cdots \mathbf{Q}_1$, and the corrupted distribution at time $t$ factorizes independently across all $L$ token positions of the action chunk:

$$q(\mathbf{a}_t \mid \mathbf{a}_0) = \prod_{i=1}^{L} \text{Categorical}(a_{t,i} \mid \bar{\mathbf{Q}}_t \, \mathbf{e}_{a_{0,i}}), \tag{2}$$

The **reverse** (denoising) process defines conditionals $p_\theta(\mathbf{a}_{t-1} \mid \mathbf{a}_t, \mathbf{c})$ under multimodal (*e.g.*, vision and language) context $\mathbf{c}$. By Bayes' rule, for each position $i$,

$$p_\theta(a_{t-1,i} \mid a_{t,i}, \mathbf{c}) \propto q(a_{t,i} \mid a_{t-1,i}) \, p_\theta(a_{t-1,i} \mid \mathbf{c}), \quad (3)$$

which, under the masking corruption $q$ in Eq. 2, reduces to

$$p_\theta(a_{t-1,i} \mid a_{t,i}, \mathbf{c}) = \begin{cases} \delta(a_{t-1,i} = a_{t,i}), & a_{t,i} \neq \mathrm{M}, \\ \mathrm{Categorical}(a_{t-1,i} \mid \pi_\theta(i \mid \mathbf{c})), & a_{t,i} = \mathrm{M}, \end{cases}$$
$$(4)$$

where $\pi_\theta(i \mid \mathbf{c})$ is the model's predictive distribution. Thus, at each step, Discrete Diffusion VLA recovers only a subset of masked positions and leaves the rest masked, denoising from higher to lower mask ratios until reaching $\mathbf{a}_0$.

In implementation, we follow mask diffusion formulations and collapse the multi-step chain into a single masked-token prediction objective. We draw a mask ratio $\gamma_t \in (0, 1]$ that emulates diffusion time $t$, replace the selected action positions by special token [MASK] to obtain $\tilde{\mathbf{a}}_t$, and train a transformer $f_\theta$ to predict the original tokens with cross-entropy on masked indices:

$$\mathcal{L}_{\mathrm{CE}}(\theta) = -\sum_{i \in \mathcal{M}_{\gamma_t}} \log p_\theta(a_{0,i} \mid \tilde{\mathbf{a}}_t, \mathbf{c}),$$
$$p_\theta(\cdot) = \mathrm{softmax}(W f_\theta(\tilde{\mathbf{a}}_t, \mathbf{c})),$$
$$(5)$$

where $\mathcal{M}_{\gamma_t}$ is the masked set and $W$ projects hidden states of action positions to the $K$-way action vocabulary. This preserves diffusion's corruption-denoising spirit while using a simple maximum-likelihood surrogate, and as shown in recent analyses (Shi et al., 2024a; Kim et al., 2025a), such losses upper-bound the negative log-likelihood under appropriate schedules.

Discrete Diffusion VLA accepts increased training-time complexity (*i.e.* solving exponentially many infilling tasks) to gain *arbitrary-order* decoding at test time, selecting the inference order adaptively, different from parallel decoding which uses a fixed, small mask ratio in a single pass and lacks a principled generative reverse chain.

### 3.3. Architecture of Discrete Diffusion VLA

**Tokenization and action chunking.** Following RT series and OpenVLA (Brohan et al., 2023; Kim et al., 2024), we discretize each control dimension via a 256-bin quantile-based scheme (1st-99th percentiles to avoid outliers), with the gripper state handled as a separate binary token. A single-timestep action thus yields $D_{\mathrm{act}} = 7$ tokens (3 translation, 3 rotation, 1 gripper), and $H$ consecutive timesteps are arranged into a fixed-length chunk of $L = H \times D_{\mathrm{act}}$ tokens, naturally suited to discrete diffusion's block-wise parallel generation paradigm.

Visual inputs comprise a mandatory third-person (head) image and two optional wrist views (Fig. 2), encoded by

SigLIP and DINOv2 and projected into the Llama2 embedding space. When provided, proprioceptive states are encoded by a lightweight MLP and concatenated with the visual tokens before being fed to the backbone.

**Unified Backbone.** We build upon OpenVLA (Kim et al., 2024), a Prismatic-7B VLM (Karamcheti et al., 2024) with a Llama 2 backbone (Touvron et al., 2023a). Unlike the original autoregressive action head, Discrete Diffusion VLA replaces causal attention over action tokens with *bidirectional* attention, allowing each action position to attend to all vision, language, and action tokens for full modality fusion. All tokens pass through the unified transformer, with hidden states at action positions projected to a 256-way vocabulary via a shared classification head. This design preserves the backbone's vision-language representation power while enabling action parallel decoding, with adaptive re-masking (Sec. 3.5) further refining uncertain tokens.

### 3.4. Algorithmic Pipeline

**Training pipeline.** For each minibatch, we sample a mask ratio $\gamma \in (0, 1]$ from a schedule (*e.g.*, cosine) emulating diffusion time, replace $\gamma L$ action positions with [MASK], and minimize masked cross-entropy following Eq. 5 with hard-label supervision, representing ground truth as one-hot vectors at the masked indices. Vision and language tokens are attended but excluded from loss. This objective is compatible with standard LM training, helping preserve pretrained VLM capabilities while aligning action generation with discrete diffusion via mask schedules. Fixed-length action chunking ($L = H \times D$ tokens) requires no [EOS] padding. The training reduces to standard cross-entropy classification at masked action positions only, with ground-truth token indices as supervision, optimizing all $L$ positions jointly in a single forward pass. No additional loss terms, auxiliary objectives, or special training procedures are involved. All parameters, including the VLM backbone and the action projection head, are updated end-to-end.

**Inference pipeline.** A key distinction from parallel decoding methods such as OpenVLA-OFT, which decode all action tokens simultaneously in a single forward pass as $\hat{\mathbf{a}}_0 = \arg\max p_\theta(\mathbf{a}_0 \mid [\text{MASK}]^L, \mathbf{c})$, is that Discrete Diffusion VLA iterates over $T$ refinement steps. At each step $t$, the model commits only the top $(1 - \gamma_t)L$ most confident positions and keeps the rest masked for subsequent refinement: For $t = T, \ldots, 1$,

$$a_{t-1,i} = \begin{cases} \text{sample } p_\theta(\cdot \mid \mathbf{a}_t, \mathbf{c})_i & \text{if } s_{t,i} \text{ is in top } (1 - \gamma_t)L, \\ [\text{MASK}] & \text{otherwise} \end{cases},$$
$$(6)$$

where $s_{t,i} = \max_k p_\theta(k \mid \mathbf{a}_t, \mathbf{c})$ is the per-position confidence. Concretely, we initialize all $L$ action positions as [MASK] and perform $T$ parallel refinement rounds, drawing candidates via multinomial sampling with a decaying tem-

perature. This makes the decoding order *adaptive* to each instance. The reverse-step conditionals follow Eq. 4, and a lightweight secondary re-masking mechanism (Sec. 3.5) applies a threshold check to prevent error propagation.

### 3.5. Adaptive Decoding and Secondary Re-Masking

**Adaptive Decoding Mechanism.** As illustrated above, the inference pipeline starts from a fully masked action chunk $\mathbf{a}_1 = \mathrm{M}^L$ with mask ratio $\gamma_1 = 1$, and then performs $T$ refinement steps with a monotone schedule $\gamma_{t+1} < \gamma_t$. Here, we use cosine schedule. At step $t$, the model yields per-position posteriors $p_\theta(\mathbf{a}_{t-1} \mid \mathbf{a}_t, \mathbf{c})$ instantiating Eq. 4. We score each position $i$ using one of the *adaptive* metrics:

$$s_{t,i} = \max_k p_\theta(k \mid \mathbf{a}_t, \mathbf{c}) \quad \textbf{(Max Confidence)}, \quad (7)$$

$$g_{t,i} = p_\theta(k_{(1)} \mid \cdot) - p_\theta(k_{(2)} \mid \cdot) \quad \textbf{(Confidence Gap)}, \quad (8)$$

with $k_{(1)}, k_{(2)}$ the labels corresponding to the highest and second-highest probabilities. Let $m_{t,i} \in \{s_{t,i}, g_{t,i}\}$. We keep the top $(1 - \gamma_{t+1})L$ positions $\mathcal{K}_t$ and update the tokens via tempered Gumbel sampling to encourage exploration:

$$a_{t+1, i \in \mathcal{K}_t} \sim \mathrm{Categorical}\left(\mathrm{softmax}\left(\frac{\log p_\theta(\cdot \mid \mathbf{a}_t, \mathbf{c}) + \varepsilon}{\tau_t}\right)\right), \quad (9)$$

where $\varepsilon$ has i.i.d. $\mathrm{Gumbel}(0, 1)$ components (equivalently, Gumbel–Max), and $\tau_t$ is a temperature that decays with $\gamma_t$. Positions not in $\mathcal{K}_t$ are set to [MASK], and the process iterates until $\gamma_T = 0$ or convergence. This instance-wise ranking makes the decoding order *adaptive* rather than fixed.

**Secondary Re-Masking.** Beyond meeting the target ratio $\gamma_{t+1}$, we run another lightweight check on previously committed tokens to prevent early errors from persisting.

*Threshold check.* If the current confidence falls below a monotonically-increasing step-dependent threshold $\eta_t^{\mathrm{abs}}$, the token is re-masked:

$$\mathcal{R}_t^{\mathrm{abs}} = \left\{ i \in \mathcal{K}_t : s_{t,i} < \eta_t^{\mathrm{abs}} \right\}. \quad (10)$$

For $i \in \mathcal{R}_t^{\mathrm{abs}}$ we set $a_{t+1, i} =$ [MASK] before proceeding to step $t + 1$. These operations maintain alignment with the Bayes reverse kernel (Eq. 4) while improving cross-iteration consistency.

## 4. Experiments

### 4.1. Simulation Benchmarks and Baselines

**Benchmarks.** We evaluate Discrete Diffusion VLA on three different robot settings: (i) Franka Panda arm on LIBERO (Liu et al., 2023) (four suites: Spatial, Object, Goal, Long; 10 tasks and 500 demos per suite); (ii) Google Robot on SimplerEnv-Fractal (Li et al., 2025), reporting *Visual Matching* and *Variant Aggregation* scores; (iii) Wid-

owX Robot on SimplerEnv-Bridge, a real-to-sim evaluation aligned with BridgeData-V2 (Walke et al., 2023).

Policies receive RGB images, language instructions, and optional end-effector positions as proprioception. No depth, affordances, or other auxiliary inputs are used. Specifically, LIBERO provides one third-person and one wrist view, while SimplerEnv provides a single third-person view.

**Baselines.** We compare against autoregressive (AR) token decoders, separate MLP action decoders, and continuous diffusion/flow-matching heads, covering both from-scratch and fine-tuned models.

*Discretized action methods:* RT-1-X/RT-2-X (O'Neill et al., 2024), OpenVLA (Kim et al., 2024), Octo-Small/Base (Ghosh et al., 2024), HPT (Wang et al., 2024), TraceVLA (Zheng et al., 2025b), SpatialVLA (Qu et al., 2025), OpenVLA-OFT (Discrete) (Kim et al., 2025b), and $\pi_0$-FAST (Pertsch et al., 2025) represent AR or PD-style generation of discrete action tokens with a unified VLM backbone or with a separate MLP decoder.

*Continuous diffusion/flow-matching methods:* Diffusion Policy (Chi et al., 2023), MDT (Reuss et al., 2024), DiT Policy (Hou et al., 2025), RoboVLM (Li et al., 2024b), $\pi_0$ (Black et al., 2024), OpenVLA-OFT (L1 loss), OpenVLA-OFT (Diffusion) (Kim et al., 2025b), GR00T-N1 (Bjorck et al., 2025), and Seer (Tian et al., 2025).

All baseline results are cited from the original publication or reproduced under identical input modalities, backbone initialization, and training budget. **A complete per-table breakdown of sources, hardware, and training steps is provided in Appendix C.**

**Training details.** We fine-tune Discrete Diffusion VLA from OpenVLA backbone (Prismatic–7B) with images resized to $224 \times 224$. For LIBERO, we train a separate policy per suite, filtering unsuccessful episodes as in Kim et al. (2025b). For SimplerEnv, we fine-tune on Fractal (Brohan et al., 2023) and BridgeData-V2 (Walke et al., 2023), respectively. Chunk sizes follow standard settings: 8 for LIBERO and SimplerEnv-Fractal, 3 for SimplerEnv-Bridge. Inference uses $T = 12$ refinement rounds with a cosine mask schedule shown most effective in Chang et al. (2022b). Details of implementation are provided in Appendix B.

### 4.2. Overall Performance and Evaluation on Vision-Language Retention

We evaluate Discrete Diffusion VLA on LIBERO for both task success and retention of pretrained vision-language capabilities. Beyond standard in-distribution (ID) evaluation, we assess out-of-distribution (OOD) generalization under two perturbation axes following LIBERO-PRO (Zhou et al., 2025): **Language Augmentation**, which paraphrases task instructions (*e.g.*, "open the top drawer and put the bowl

*Table 1.* **LIBERO task performance results (%).** Each column is a LIBERO task suite; values are averaged over 500 rollouts per suite (10 tasks × 50 episodes). Methods above the horizontal rule are continuous action methods; **those below are discretized action methods**. Best are bold and best in each part are underlined. We report the highest score across seeds.

| Model | Spatial | Object | Goal | Long | Average |
|---|---|---|---|---|---|
| Diffusion Policy (scratch) | 78.3 | 92.5 | 68.3 | 50.5 | 72.4 |
| MDT (scratch) | 78.5 | 87.5 | 73.5 | 64.8 | 76.1 |
| Seer (scratch) | – | – | – | 78.7 | – |
| Seer (fine-tuned) | – | – | – | 87.7 | – |
| Dita / DiT Policy | 84.2 | 96.3 | 85.4 | 63.8 | 82.4 |
| $\pi_0$ | 96.8 | 98.8 | 95.8 | 85.2 | 94.2 |
| OpenVLA-OFT (Diffusion) | 96.9 | 98.1 | 96.2 | 91.1 | 95.6 |
| OpenVLA-OFT (L1) | **97.6** | 98.4 | **97.9** | **94.5** | **97.1** |
| GR00T-N1 | 94.4 | 97.6 | 93.0 | 90.6 | 93.9 |
| OpenVLA | 84.7 | 88.4 | 79.2 | 53.7 | 76.5 |
| Octo-Base | 78.9 | 85.7 | 84.6 | 51.1 | 75.1 |
| TraceVLA | 84.6 | 85.2 | 75.1 | 54.1 | 74.8 |
| SpatialVLA | 88.2 | 89.9 | 78.6 | 55.5 | 78.1 |
| $\pi_0$ + FAST | 96.4 | 96.8 | 88.6 | 60.2 | 85.5 |
| OpenVLA-OFT (Discrete) | 96.2 | 98.2 | 95.6 | 92.0 | 95.5 |
| Discrete Diffusion VLA | 97.2 | **99.4** | 96.8 | 92.2 | 96.4 |

inside" → "slice open the top drawer and place the bowl in it"), and **Vision Augmentation**, which alters object appearances including materials, colors, and sizes (Fig. 3). This joint evaluation directly tests whether our unified architecture preserves VLM priors while achieving strong action decoding. Full OOD settings and all augmented instructions are detailed in Appendix D.

**In-distribution results.** Table 1 reports success rates on four LIBERO suites. Discrete Diffusion VLA achieves 96.4% average SR with per-suite scores of 97.2% (Spatial), 99.4% (Object), 96.8% (Goal), and 92.2% (Long), establishing state-of-the-art among all discrete tokenized methods (+0.9% over OpenVLA-OFT Discrete). Against methods trained from scratch, Discrete Diffusion VLA surpasses Diffusion Policy and MDT by +24.0 and +20.3 points respectively. The overall continuous SOTA, OpenVLA-OFT (L1) at 97.1%, uses continuous action representations that inherently avoid quantization error. Despite this inherent disadvantage, Discrete Diffusion VLA narrows the gap to only 0.7% on LIBERO, achieves state-of-the-art across all methods on SimplerEnv (Sec. 4.3), and offers substantially stronger OOD generalization as illustrated below.

**Out-of-distribution results.** Tables 2 and 3 reveal a key advantage of our unified architecture in preserving vision-language capabilities under distribution shift. All results are averaged over 500 rollouts per suite. On LIBERO-Goal-OOD, Discrete Diffusion VLA exhibits only 0.8% language degradation, compared to 8.0% for parallel decoding, 2.4% for continuous diffusion, and 3.2% for L1 regression. Vision degradation is similarly reduced at 20.4%, against 22.6%, 29.0%, and 23.2% respectively. Notably, while OpenVLA-

OFT (L1) achieves the highest in-distribution (ID) accuracy, Discrete Diffusion VLA attains the best absolute OOD performance with the smallest degradation under both perturbations, a pattern that holds consistently on LIBERO-Spatial as well. Continuous diffusion suffers the most severe vision degradation (↓29.0%), consistent with previous findings in Yang et al. (2026b); Liu et al. (2025) that separate diffusion heads become overly vision-dependent.

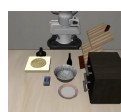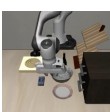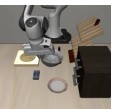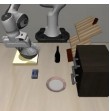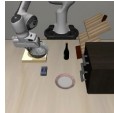

*Figure 3.* **Vision Augmentation Sample of LIBERO-OOD Goal Task.** OOD scenes include objects with different scale, materials, and appearance (*e.g.*, larger bowl, metallic stove).

*Table 2.* **Out-of-distribution performance on LIBERO-Goal**

| Method | Original | Lang Aug | Vision Aug |
|---|---|---|---|
| OpenVLA-OFT (Discrete) | 95.6% | 87.6% (↓8.0%) | 73.0% (↓22.6%) |
| OpenVLA-OFT (Diffusion) | 96.0% | 93.6% (↓2.4%) | 67.0% (↓29.0%) |
| OpenVLA-OFT (L1) | **97.9%** | 94.7% (↓3.2%) | 74.7% (↓23.2%) |
| **Discrete Diffusion VLA** | 96.8% | **96.0%** (↓**0.8%**) | **76.4%** (↓**20.4%**) |

*Table 3.* **Out-of-distribution performance on LIBERO-Spatial**

| Method | Original | Lang Aug | Vision Aug |
|---|---|---|---|
| OpenVLA-OFT (Discrete) | 96.2% | 94.6% (↓1.6%) | 95.0% (↓1.2%) |
| OpenVLA-OFT (Diffusion) | 96.9% | 94.3% (↓2.6%) | 91.1% (↓5.8%) |
| OpenVLA-OFT (L1) | **97.6%** | **96.2%** (↓1.4%) | 96.0% (↓1.6%) |
| **Discrete Diffusion VLA** | 97.2% | **96.2%** (↓**1.0%**) | **96.4%** (↓**0.8%**) |

This advantage stems from training on the same cross-entropy objective over the same token space as the pre-trained VLM, preserving priors by construction rather than by regularization. Meanwhile, bidirectional attention over action chunks and confidence-based decoding order yield consistent gains over AR decoding as shown above. We discuss the complementary advantages over both paradigms in Appendix E.

### 4.3. Extended Evaluation Across Robot Platforms

**Google Robot (SimplerEnv-Fractal).** As shown in Tab. 4, Discrete Diffusion VLA achieves state-of-the-art performance across both discrete and continuous methods. On *Visual Matching*, we obtain 71.2% average sucess rates, surpassing $\pi_0$ (58.8%), $\pi_0$-FAST (61.9%), and OpenVLA-OFT (63.0%). On *Variant Aggregation*, Discrete Diffusion VLA attains 56.9%, competitive with RT-2-X (64.3%) and $\pi_0$-FAST (59.0%). Aggregating both metrics, ours yields the highest overall average of 64.1%.

**WidowX Robot (SimplerEnv-Bridge).** Tab. 5 shows Discrete Diffusion VLA achieves SOTA performance with 54.2% overall, outperforming all continuous diffusion/flow-matching policies ($\pi_0$: 40.1%, +14.1%; GR00T-N1: 49.5%, +4.7%) and discrete baselines ($\pi_0$-FAST: 48.3%, +5.9%).

*Table 4.* **SimplerEnv evaluation across different policies on Google Robot tasks**. We report the results of all models pretrained with OXE dataset (O'Neill et al., 2024) and then fine-tuned with Fractal dataset (Brohan et al., 2023).

| Model | Visual Matching | | | | Variant Aggregation | | | | #Overall |
|---|---|---|---|---|---|---|---|---|---|
| | Pick Coke | Mv Near | Drawer | Avg. | Pick Coke | Mv Near | Drawer | Avg. | Average |
| RT-1-X (O'Neill et al., 2024) | 56.7% | 31.7% | 59.7% | 53.4% | 49.0% | 32.3% | 29.4% | 39.6% | 46.5% |
| RT-2-X (O'Neill et al., 2024) | 78.7% | **77.9%** | 25.0% | 60.7% | 82.3% | **79.2%** | **35.3%** | **64.3%** | 62.5% |
| Octo-Base (Ghosh et al., 2024) | 17.0% | 4.2% | 22.7% | 16.8% | 0.6% | 3.1% | 1.1% | 1.1% | 9.0% |
| OpenVLA (Kim et al., 2024) | 16.3% | 46.2% | 35.6% | 27.7% | 54.5% | 47.7% | 17.7% | 39.8% | 33.8% |
| HPT (Wang et al., 2024) | 56.0% | 60.0% | 24.0% | 46.0% | – | – | – | – | – |
| Moto (Chen et al., 2025) | 74.0% | 60.4% | 43.1% | 59.2% | – | – | – | – | – |
| RoboVLM (Li et al., 2024b) | 77.3% | 61.7% | 43.5% | 63.4% | 75.6% | 60.0% | 10.6% | 51.3% | 57.4% |
| TraceVLA (Zheng et al., 2025b) | 28.0% | 53.7% | 57.0% | 42.0% | 60.0% | 56.4% | 31.0% | 45.0% | 43.5% |
| $\pi_0$ (Black et al., 2024) | 72.7% | 65.3% | 38.3% | 58.8% | 75.2% | 63.7% | 25.6% | 54.8% | 56.8% |
| $\pi_0$-FAST (Pertsch et al., 2025) | 75.3% | 67.5% | 42.9% | 61.9% | 77.6% | 68.2% | 31.3% | 59.0% | 60.5% |
| OpenVLA-OFT (Kim et al., 2025b) | 72.3% | 69.6% | 47.2% | 63.0% | 65.3% | 59.0% | 12.2% | 45.5% | 54.3% |
| GR00T-N1 (Bjorck et al., 2025) | 47.0% | 70.0% | 18.1% | 45.0% | 78.8% | 62.5% | 13.2% | 51.5% | 48.4% |
| Discrete Diffusion VLA | **85.4%** | 67.5% | **60.6%** | **71.2%** | 82.5% | 64.6% | 23.6% | 56.9% | **64.1%** |

*Table 5.* **SimplerEnv evaluation across different policies on WidowX Robot tasks**. We report the results of all models pretrained with OXE dataset (O'Neill et al., 2024) and then fine-tuned with BridgeData V2 (Walke et al., 2023).

| Method | Put Spoon on Towel | | Put Carrot on Plate | | Stack Green on Yellow | | Put Eggplant in Basket | | #Overall |
|---|---|---|---|---|---|---|---|---|---|
| | Grasp Spoon | Success | Grasp Carrot | Success | Grasp G Block | Success | Grasp Eggplant | Success | Average |
| RT-1-X (O'Neill et al., 2024) | 16.7% | 0.0% | 20.8% | 4.2% | 8.3% | 0.0% | 0.0% | 0.0% | 6.3% |
| Octo-Base (Ghosh et al., 2024) | 34.7% | 12.5% | 52.8% | 8.3% | 31.9% | 0.0% | 66.7% | 43.1% | 31.3% |
| Octo-Small (Ghosh et al., 2024) | 77.8% | 47.2% | 27.8% | 9.7% | 40.3% | 4.2% | 87.5% | 56.9% | 43.9% |
| OpenVLA (Kim et al., 2024) | 4.1% | 0.0% | 33.0% | 0.0% | 12.5% | 0.0% | 8.3% | 4.1% | 7.8% |
| RoboVLM (Li et al., 2024b) | 54.2% | 29.2% | 25.0% | 25.0% | 45.8% | 12.5% | 58.3% | 58.3% | 38.5% |
| $\pi_0$ (Black et al., 2024) | 45.8% | 29.1% | 25.0% | 0.0% | 50.0% | 16.7% | 91.6% | 62.5% | 40.1% |
| $\pi_0$-FAST (Pertsch et al., 2025) | 62.5% | 29.1% | **58.5%** | 21.9% | 54.0% | 10.8% | 83.3% | 66.6% | 48.3% |
| OpenVLA-OFT (Kim et al., 2025b) | 50.0% | 12.5% | 41.7% | 4.2% | **70.8%** | 8.3% | **91.7%** | 37.5% | 39.6% |
| GR00T-N1 (Bjorck et al., 2025) | **83.3%** | **62.5%** | 54.2% | **45.8%** | 70.8% | 16.7% | 41.7% | 20.8% | 49.5% |
| Discrete Diffusion VLA | 70.8% | 29.2% | 58.3% | 29.2% | 62.5% | **20.8%** | 91.7% | 70.8% | **54.2%** |

Per-task breakdown shows consistent gains in both grasp and success metrics (*e.g.*, Put Eggplant in Basket: 91.7% grasp / 70.8% success).

Across both SimplerEnv benchmarks, Discrete Diffusion VLA achieves best performance regardless of action representation, demonstrating discrete diffusion inside a unified transformer can match and has potential to exceed performance of other baselines while preserving VLM priors.

### 4.4. Ablation Study

**Action head without robot pretraining.** To verify that the gains of discrete diffusion are not artifacts of OpenVLA pretraining, we ablate action head variants on a pure VLM backbone (Qwen2.5-VL) with no robot-specific initialization. Discrete diffusion achieves the best average performance across all LIBERO suites, outperforming AR, FAST, parallel decoding, and continuous diffusion, confirming that the advantage is intrinsic to the discrete diffusion paradigm. Full results are provided in Appendix F.

**Decoding strategy.** We compare one-shot parallel decoding, random order, confidence-gap selection, and our max-confidence selection. On LIBERO-Goal, success rates are 95.6%, 95.8%, 96.6%, and 96.8% respectively (Tab. 7), a +1.2% gain from adaptive decoding over one-shot parallel.

*Table 6.* **Inference speed comparison on LIBERO-Goal.**

| Method | Latency (ms) | Speed (Hz) | NFE |
|---|---|---|---|
| OpenVLA (AR) | 136.2 | 7.34 | 56 |
| OpenVLA w/o KVcache (AR) | 209.5 | 4.77 | 56 |
| OpenVLA-OFT (Parallel Decoding) | 31.1 | 32.14 | 1 |
| OpenVLA-OFT (Diffusion, 50 steps) | 199.9 | 5.00 | 50 |
| OpenVLA-OFT (Diffusion, 12 steps) | 67.1 | 14.91 | 12 |
| **Discrete Diffusion VLA (12 steps)** | **68.8** | **14.53** | **12** |

Max-confidence slightly outperforms confidence-gap, while random order lags behind, confirming that instance-wise confidence ranking with high-confidence-first resolution improves refinement effectiveness.

**Choice temperature.** Tab. 8 shows that linear decay from 1.0 to 0.0 achieves 96.8%, outperforming hard argmax (96.2%) and fixed temperature (96.4%). Decay encourages mild exploration early and sharper commitment later, complementing adaptive decoding to correct early mistakes and consolidate consistent actions.

**Denoising steps and speed–quality trade-off.** Fig. 4 sweeps $T$ and reports throughput (number of chunks per second) and success rates on LIBERO-Goal. Accuracy generally improves with $T$ while efficiency scales inversely. We adopt $T=12$ as a practical operating point that achieves strong accuracy without incurring prohibitive latency.

*Table 7.* **Ablation study on decoding strategy.** (LIBERO-Goal). Ranking tokens by instance-wise confidence improves over one-shot parallel, and our max confidence yields the best accuracy (96.8%).

| Decoding Strategy | Parallel Decoding | +Random Order | +Confidence Gap | +Max Confidence |
|---|---|---|---|---|
| **Success** | 95.6% | 95.8% | 96.6% | **96.8%** |

*Table 8.* **Ablation on choice temperature.** (LIBERO-Goal). Linear decay temperature attains the best 96.8%, encouraging mild exploration early and sharper commitment later.

| Choice Temperature | Hard Sample (Temp=0) | Fixed Temp (Temp=1) | Linear Decay Temp (Temp=1-$t$) |
|---|---|---|---|
| **Success Rates** | 96.2% | 96.4% | **96.8%** |

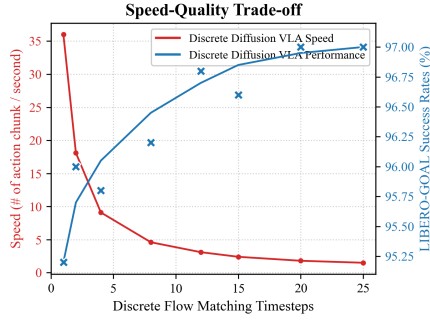

*Figure 4.* **Speed–Quality trade-off.** (i) Left y-axis: Time efficiency by the number of generated chunks per second. (ii) Right y-axis: Ablation on denoising steps.

## 4.5. Inference Efficiency

Discrete diffusion offers efficiency advantages over AR decoding in both number of function evaluations (NFEs) and wall-clock latency. For an action chunk of length $L = H \times D_{act}$, AR requires $L$ sequential forward passes. In LIBERO ($H=8$, $D_{act}=7$), this yields $L=56$ passes, a latency bottleneck scaling linearly with horizon. Discrete Diffusion VLA denoises the entire chunk in $T$ steps, where each step is a single forward pass predicting posteriors for all currently masked tokens. With $T=12$, NFEs drop from 56 to 12 (4.7× fewer), decoupling inference cost from sequence length. The adaptive decoding and secondary remasking operate on logits and add no extra forward passes, preserving efficiency of parallel refinement.

Table 6 reports end-to-end latency on a single NVIDIA H800 GPU. Discrete Diffusion VLA achieves 68.8 ms per chunk (14.53 Hz), 2× faster than AR (136.2 ms), and comparable to continuous diffusion when using same denoising steps under advanced optimizer (67.1 ms). While one-shot parallel decoding is fastest (31.1 ms), it sacrifices accuracy (*e.g.*, 54.3% vs. 64.1% on SimplerEnv-Fractal). Secondary remasking adds negligible overhead (<1 ms) as it operates purely on logits.

## 4.6. Real-Robot Evaluation

To validate real-world practicality, we deploy Discrete Diffusion VLA on an AgileX Cobot Magic dual-arm platform (ALOHA-style) across two tabletop manipulation tasks (Fig. 5): *click the bell* and *place cup on coaster*. We first collect 150 demonstrations in RoboTwin simulation for domain alignment (80k steps), then fine-tune on 150 real-robot demonstrations (200k steps), with an action chunk of $H=12$. Each task is evaluated over 15 trials.

Table 9 shows that Discrete Diffusion VLA outperforms both OpenVLA-OFT and $\pi_0$ on *click the bell*, and matches $\pi_0$ on *place cup on coaster*. The control frequency of 9.69 Hz on a single RTX 4090 is sufficient for quasi-static tabletop manipulation, while the higher frequency of $\pi_0$

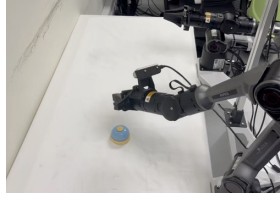 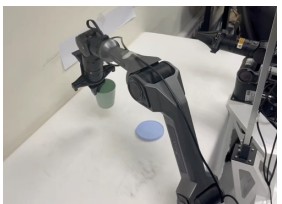

| (a) click the bell | (b) place cup on coaster |
|---|---|

*Figure 5.* **Real-robot task setups on AgileX Cobot Magic.** Evaluated over 15 trials per task.

*Table 9.* **Real-robot evaluation on AgileX Cobot Magic.**

| Method | Click Bell | Place Cup | Latency (Hz) |
|---|---|---|---|
| OpenVLA-OFT (Discrete) | 33.3% | 20.0% | 34.3 |
| $\pi_0$ | 53.3% | **40.0%** | 24.5 |
| **Discrete Diffusion VLA** | **66.7%** | **40.0%** | 9.69 |

reflects engineering optimization during deployment rather than architectural advantages. These results confirm that discrete diffusion transfers effectively to real hardware. Visualizations are available in Appendix A.

## 4.7. Visualization of Adaptive Decoding Order

Fig. 6 visualizes the adaptive decoding order of Discrete Diffusion VLA across 12 denoising steps. Each grid represents an action chunk as an $8 \times 7$ matrix, where rows correspond to 8 sequential actions in a chunk and columns to 7-dimensional action tokens. A token's decoding difficulty is shaped by multiple factors including occurrence frequency, task structure, and the context provided by already-demasked tokens. Among these, training frequency serves as the most accessible and informative proxy, as tokens appearing more frequently tend to be learned more robustly. We therefore encode frequency as color intensity, with darker brown/orange shades indicating higher-frequency tokens and light gray indicating masked tokens.

High-frequency tokens predominantly appear in early steps ($t=0 \sim 6$), supporting the intuition that easier elements are resolved first. Notably, some low-frequency tokens are demasked early when contextually predictable, confirming

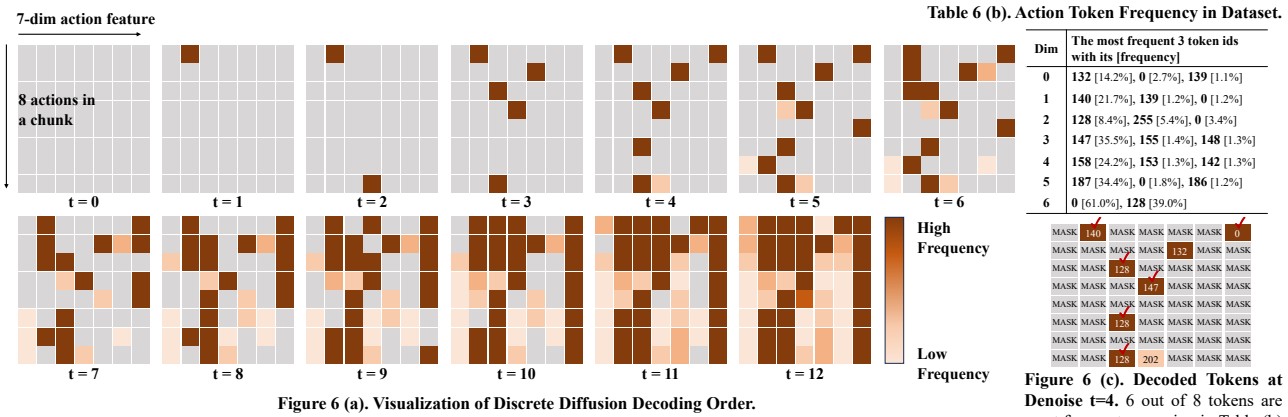

**Table 6 (b). Action Token Frequency in Dataset.**

| Dim | The most frequent 3 token ids with its [frequency] |
|---|---|
| 0 | **132** [14.2%], **0** [2.7%], **139** [1.1%] |
| 1 | **140** [21.7%], **139** [1.2%], **0** [1.2%] |
| 2 | **128** [8.4%], **255** [5.4%], **0** [3.4%] |
| 3 | **147** [35.5%], **155** [1.4%], **148** [1.3%] |
| 4 | **158** [24.2%], **153** [1.3%], **142** [1.3%] |
| 5 | **187** [34.4%], **0** [1.8%], **186** [1.2%] |
| 6 | **0** [61.0%], **128** [39.0%] |

**Figure 6 (c). Decoded Tokens at Denoise t=4.** 6 out of 8 tokens are most frequent appearing in Table (b).

Figure 6 (a). Visualization of Discrete Diffusion Decoding Order.

*Figure 6.* **Visualization and Analysis of Adaptive Decoding Order**

that the adaptive strategy responds to instance-wise confidence shaped by all factors rather than dataset statistics alone. Fig. 6 (c) quantifies this at $t=4$: 6 out of 8 decoded tokens belong to the top-3 most frequent tokens per dimension (Tab. 6 (b)), providing qualitative evidence for the high-confidence-first decoding strategy.

## 5. Conclusion

We present Discrete Diffusion VLA, a VLA that unifies perception, language grounding, and action generation within a single transformer through discrete diffusion over action tokens. Sharing the same cross-entropy objective and token vocabulary as the pretrained VLM preserves vision-language priors by construction, avoiding the gradient competition that erodes OOD robustness in continuous diffusion heads. Compared to autoregressive decoding, discrete diffusion enables bidirectional attention over the full action chunk and eliminates compounding errors through confidence-guided adaptive decoding. Experiments across simulation benchmarks and a real robot platform confirm state-of-the-art performance among discrete methods, superior generalization under distribution shift, and favorable inference efficiency over autoregressive decoding.

**Limitations and future work.** Our multi-step iterative decoding is slower than single-pass decoding by design, and variable-length action tokenization schemes (*e.g.* $\pi_0$-FAST) are incompatible with discrete diffusion. Adopting more expressive fixed-length tokenization strategies to further close the gap with continuous methods remains an important direction.

## Acknowledgments

We sincerely thank Wanqi Zhong from Harbin Institute of Technology for assistance with figure illustration and layout. This paper is partially supported by the National Key R&D Program of China No.2022ZD0161000, the General Research Fund of Hong Kong No.17208825, 17200622 and 17209324, and the Chinese Institute of Electronics-Tencent Doctoral Research Incentive Program.

## Impact Statement

This paper presents work whose goal is to advance the field of robot learning through vision-language-action models. While we envision positive applications in assistive and industrial robotics, deployment in safety-critical environments requires formal safety evaluation beyond the scope of this work. Additionally, the reliance on large pretrained vision-language models may inherit biases from pretraining data, which should be considered before real-world deployment.

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

# A. Visualizations of Robot Task Executions

i) Franka Panda Arm on LIBERO-Spatial Task

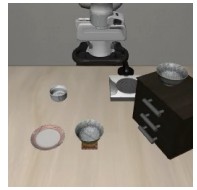 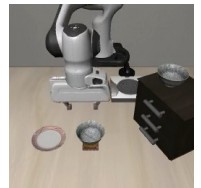 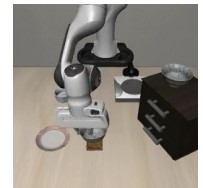 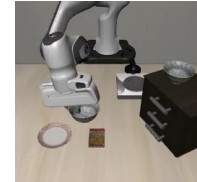 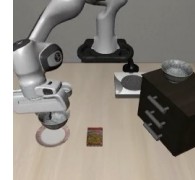 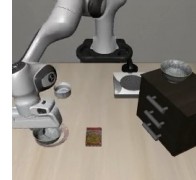

ii) Google Robot on Move Near Task

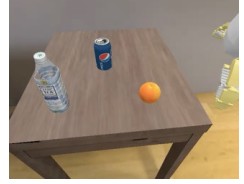 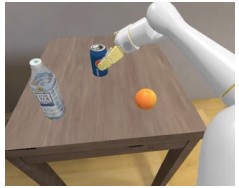 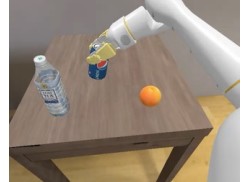 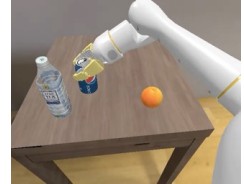 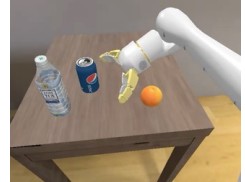

iii) WidowX Arm on Put Eggplant in Basket Task

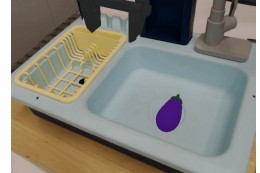 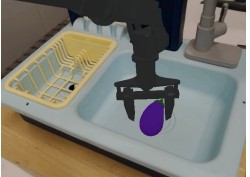 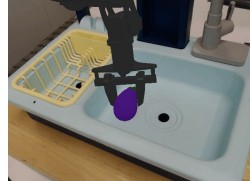 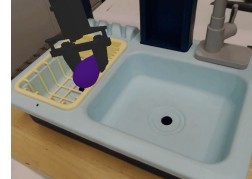 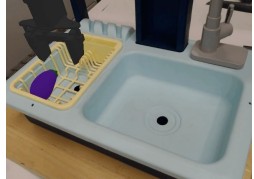

iv) Franka Panda Arm on LIBERO-Long Task

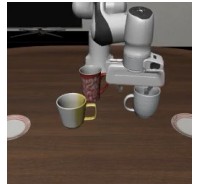 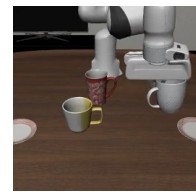 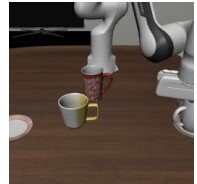 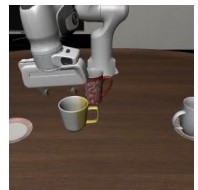 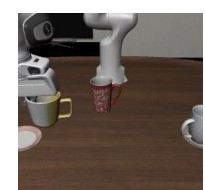 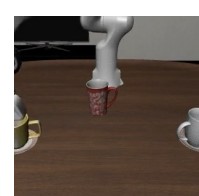

v) WidowX Arm on Stack Green Block on Yellow Block Task

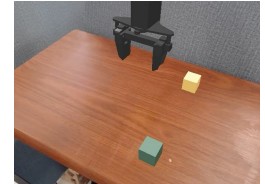 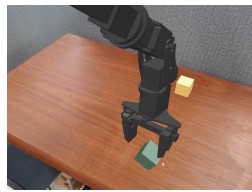 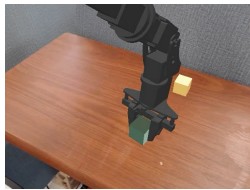 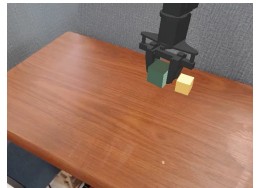 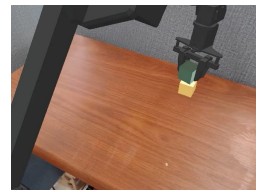

vi) AgileX Cobot Magic on Click the Bell Task

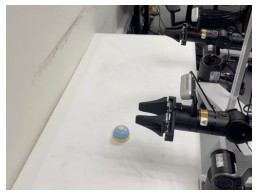 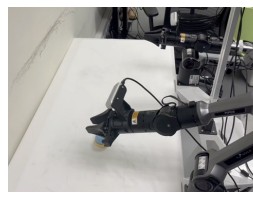 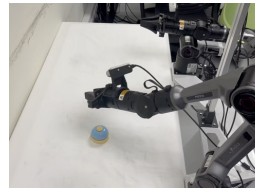 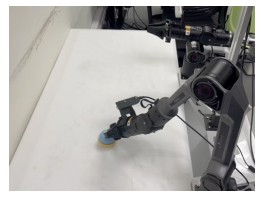 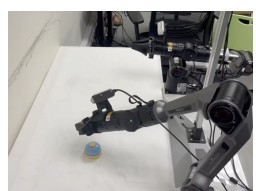

vii) AgileX Cobot Magic on Place Cup on Coaster Task

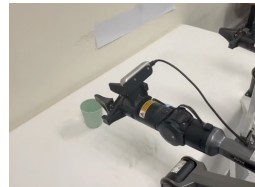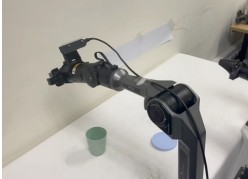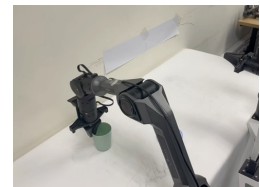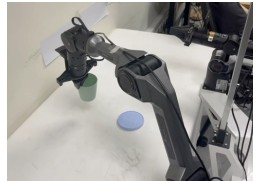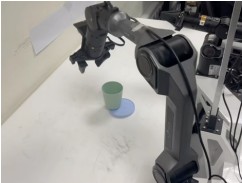

## B. Implementation Details

1. We choose action chunk size $H$ as 8 for both LIBERO and SimplerEnv-Fractal, and 3 for SimplerEnv-Bridge, following widely used settings in each environment, respectively.

2. We conduct our experiments with batch size 32 for all of the experiments and typically conduct each run on 4 NVIDIA A800 TENSOR CORE GPUs.

3. We apply Feature-wise Linear Modulation (FiLM) (Kim et al., 2025b) on Simpler-Bridge experiments to enhance the language grounding abilities of our model on WidowX Arm manipulation tasks.

4. We train our model on LIBERO-Spatial and LIBERO-Object for 150k steps while 300k steps on LIBERO-Goal and LIBERO-Long. And we report the results of each highest checkpoints. Besides, we only train 100k steps on both Simpler-Fractal and Simpler-Bridge and report the highest overall performance of each environment.

5. For secondary re-masking, we set $\eta_t^{abs} = 0.5 \times (1 - t/T)$ as the step-dependent threshold of threshold check.

## C. Reproducibility and Baseline Details

For transparency, we clarify which baseline results are taken from published papers and which are reproduced by us under controlled conditions. All reproduced baselines use the same input modalities, data splits, and evaluation protocol as Discrete Diffusion VLA. Table 10 summarizes the source of every result reported in the main paper.

*Table 10.* **Source of baseline results across all main-paper tables.** The reported batch size is the total batch size across all GPUs.

| Table | Method | Source / Training Details |
|---|---|---|
| Table 1 | All except below | Cited from OpenVLA-OFT (Kim et al., 2025b) |
| | $\pi_0$-FAST, GR00T-N1 | Cited from Hume (Song et al., 2026) |
| | OpenVLA-OFT (Discrete) | Reproduced; OpenVLA ckpt, 150k steps, 8×A100, B64 |
| Tables 2-3 | All | Reproduced; OpenVLA ckpt, 200k steps, 2×H800, B16 |
| Tables 4-5 | All except below | Cited from SpatialVLA (Qu et al., 2025) |
| | Moto | Cited from Moto (Chen et al., 2025) |
| | $\pi_0$-FAST | Cited from Hume (Song et al., 2026) |
| | GR00T-N1 | Reproduced; GR00T ckpt, 60k steps, 8×A100, B1024 |
| | OpenVLA-OFT | Reproduced; OpenVLA ckpt, 100k steps, 8×A100, B64 |
| Table 6 | All | Measured on 1×H800 |

For all reproduced baselines, we use the official pretrained checkpoints released by the respective authors and fine-tune under identical data and augmentation conditions as Discrete Diffusion VLA. No additional data filtering or pretraining budget beyond what is specified above was applied. All experiments are run with 3 random seeds and the mean is reported, consistent with the OOD evaluation protocol described in Appendix D.

# D. Out-of-Distribution Evaluation Benchmark

We follow the experimental settings of LIBERO-PRO (Zhou et al., 2025) while correcting several inconsistencies identified in the published implementation.

## D.1. Visual Augmentation

Visual augmentation follows the use_object protocol of LIBERO-PRO exactly, replacing in-distribution objects with visually distinct variants differing in scale, material, and appearance. We apply this perturbation to both LIBERO-Goal and LIBERO-Spatial.

For LIBERO-Goal, one configuration substitutes a larger bowl and a stove with metallic luster to alter both scale and surface reflectance.

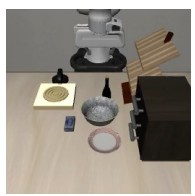 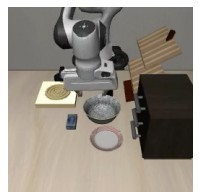 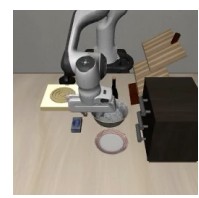 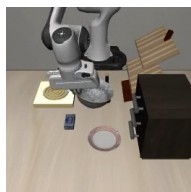 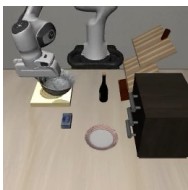 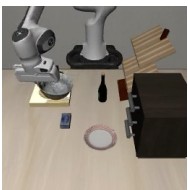

A second configuration replaces a bottle and a cabinet with variants of different materials and textures, further challenging appearance-level generalization.

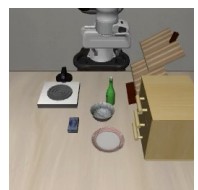 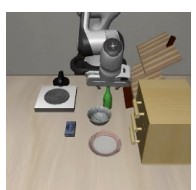 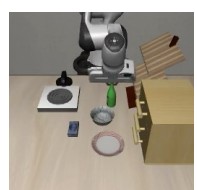 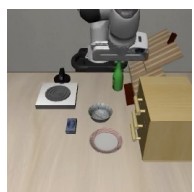 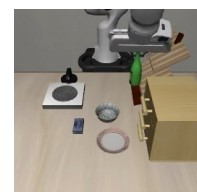 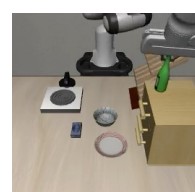

For LIBERO-Spatial, objects are similarly substituted with out-of-distribution variants across scale, material, and appearance.

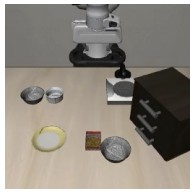 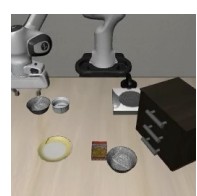 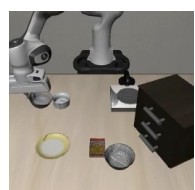 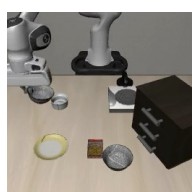 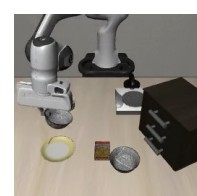 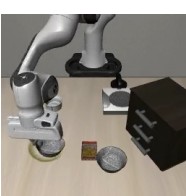

## D.2. Language Augmentation

Language augmentation corrects the use_lang protocol of LIBERO-PRO, replacing original task instructions with semantically equivalent but lexically varied alternatives to test robustness to instruction paraphrase. Table 11 lists all ten OOD instructions for both LIBERO-Goal and LIBERO-Spatial, with italic text marking modified words or phrases relative to the original in-distribution instructions.

# E. Why Discrete Diffusion: Complementary Advantages over AR and Continuous Diffusion

Discrete Diffusion VLA is not a compromise between AR decoding and continuous diffusion. It resolves the core limitation of each. AR methods that unify action generation within the VLM backbone preserve pretrained priors but suffer from causal commitment and compounding errors. Continuous diffusion heads enable structured parallel generation but erode VLM priors through competing gradients from a separate loss. Discrete diffusion inside the unified backbone achieves both prior preservation and structured parallel generation simultaneously, without either pathology. We elaborate on each dimension below.

*Table 11.* **OOD language instructions for LIBERO benchmarks.** Italic text indicates modifications from original instructions.

| # | libero_goal_lang (OOD) | libero_spatial_lang (OOD) |
|---|---|---|
| 1 | open the middle *dresser* of the cabinet | *lift* the black bowl between the plate and the ramekin and *set* it on the plate |
| 2 | *slice open* the top drawer and *place* the bowl inside | *lift* the black bowl from table center and *put* it on the plate |
| 3 | *move* the plate to the front of the stove | *take* the black bowl in the top drawer of the wooden cabinet and *put* it on the plate |
| 4 | *place* the bowl on *top of* the plate | *lift* the black bowl *beside* the cookie box and *put* it on the plate |
| 5 | *place* the bowl on the stove | *pick* the black bowl *beside* the plate and *put* it on the plate |
| 6 | *place* the bowl on top of the cabinet | *take* the black bowl next to the ramekin and *place* it on the plate |
| 7 | *place* the cream cheese in the bowl | *lift* the black bowl on the cookie box and *put* it on the plate |
| 8 | *place* the wine bottle on the rack | *grab* the black bowl on the ramekin and *set* it on the plate |
| 9 | *place* the wine bottle on top of the cabinet | *lift* the black bowl on the stove and *set* it on the plate |
| 10 | *switch* on the stove | *lift* the black bowl on the wooden cabinet and *put* it on the plate |

## E.1. Resolving the Limitations of Autoregressive Decoding

AR methods such as OpenVLA also achieve architectural unification by keeping action generation inside the same transformer with a cross-entropy objective. However, this shared property alone does not distinguish discrete diffusion from AR. The key structural difference is that discrete diffusion trains the model to solve exponentially many masked infilling tasks simultaneously, enabling bidirectional attention over the full action chunk and arbitrary-order decoding at inference, both of which are structurally unavailable to AR.

**Bidirectional attention for structured action chunk modeling.** AR commits tokens in a fixed left-to-right order under causal attention, where each position is decided without knowledge of later action dimensions. Action chunks are structured sets of mutually dependent decisions, not causal sequences where earlier tokens should be irrevocable. Discrete diffusion employs bidirectional attention over the full action chunk, letting each position attend to all visual, language, and action context simultaneously.

**Adaptive decoding order without compounding errors.** Rather than fixed left-to-right commitment, discrete diffusion first forms a coarse global hypothesis, then progressively refines low-confidence tokens with confidence-guided easy-first ordering (Fig. 6). Secondary remasking further corrects inconsistent predictions, which is impossible in standard AR where early errors propagate irreversibly. On a controlled comparison using an identical Qwen2.5-VL backbone with no robot pretraining (Appendix F), discrete diffusion achieves 95.1% versus 81.2% for AR under matched conditions.

**Inference efficiency.** AR requires $L$ sequential forward passes (NFE=56 for LIBERO), with latency scaling linearly in chunk length. Discrete Diffusion VLA decodes the full chunk in $T$=12 parallel steps at 14.53 Hz, achieving a 4.7$\times$ reduction in NFEs and a 2$\times$ wall-clock speedup over AR (Table 6).

## E.2. Resolving the Limitations of Continuous Diffusion Heads

Continuous diffusion methods introduce a diffusion loss that operates on a different objective and modality from the pretrained cross-entropy loss over language tokens. Even when the diffusion head shares the same backbone, this loss mismatch creates competing gradients that erode pretrained VLM priors, manifesting most clearly under distribution shift.

Table 12 presents OOD degradation in isolation to highlight relative robustness. Continuous diffusion suffers the most severe vision degradation ($\downarrow$29.0% on LIBERO-Goal, $\downarrow$5.8% on LIBERO-Spatial), consistent with prior findings that

diffusion-based action heads become overly vision-dependent (Yang et al., 2026b; Liu et al., 2025). Discrete Diffusion VLA avoids this by construction, as discrete action tokens share the same vocabulary space and training objective as language tokens, so no competing loss is introduced at any point. The narrower OOD gaps on LIBERO-Spatial reflect its dataset design, where same objects with varied spatial layouts naturally encourage generalization during training, making the OOD advantage more pronounced on LIBERO-Goal.

*Table 12.* **OOD degradation comparison.** Values indicate absolute performance drop from in-distribution. Discrete Diffusion VLA exhibits the smallest degradation across both benchmarks and both perturbation types.

| Method | LIBERO-Goal | | LIBERO-Spatial | |
|---|---|---|---|---|
| | **Lang Aug** | **Vision Aug** | **Lang Aug** | **Vision Aug** |
| OpenVLA-OFT (Diffusion) | ↓2.4% | ↓29.0% | ↓2.6% | ↓5.8% |
| OpenVLA-OFT (L1) | ↓3.2% | ↓23.2% | ↓1.4% | ↓1.6% |
| **Discrete Diffusion VLA** | **↓0.8%** | **↓20.4%** | **↓1.0%** | **↓0.8%** |

In summary, discrete diffusion inside a unified VLM backbone inherits the prior-preservation benefits of architectural unification from AR and the structured parallel generation benefits from diffusion, while avoiding the compounding errors of AR and the gradient competition introduced by any diffusion loss operating alongside the pretrained cross-entropy objective.

## F. Action Head Ablation without Robot Pretraining

A natural concern is whether the performance of Discrete Diffusion VLA stems from the discrete diffusion paradigm itself or is an artifact of inheriting OpenVLA pretrained weights, given the mismatch between the autoregressive pretraining objective and the discrete diffusion fine-tuning objective. To isolate the contribution of the action head, we replace the OpenVLA backbone with Qwen2.5-VL (**?**), a pure vision-language model with no robot-specific pretraining, and compare four action head variants under otherwise identical conditions: all four LIBERO suites trained jointly for 30k steps with 256-bin discretization and a learning rate of 1e-4.

*Table 13.* **Action head comparison on Qwen2.5-VL without robot pretraining.** All models are trained jointly on all four LIBERO suites from a pure VLM initialization.

| Action Head | Spatial | Object | Goal | Long | Avg |
|---|---|---|---|---|---|
| AR | 79.8 | 91.8 | 84.0 | 69.3 | 81.2 |
| Parallel Decoding | 96.0 | 97.6 | 94.2 | 89.0 | 94.2 |
| FAST | 89.0 | 97.2 | 91.2 | 87.0 | 91.1 |
| Diffusion (GR00T-style) | 95.8 | 98.2 | 94.6 | 90.8 | 94.9 |
| **Discrete Diffusion (Ours)** | **95.8** | **98.8** | **95.4** | **90.4** | **95.1** |

Table 13 shows that discrete diffusion achieves the highest average success rate (95.1%), outperforming AR by 13.9%, FAST by 4.0%, parallel decoding by 0.9%, and continuous diffusion by 0.2%. Critically, these gains are obtained on a backbone with no robot pretraining whatsoever, confirming that the performance advantage is intrinsic to the discrete diffusion paradigm rather than inherited from OpenVLA initialization. The AR head suffers most (81.2%), consistent with the well-known difficulty of fitting action distributions with a unimodal sequential prediction objective. The small but consistent margin of discrete over continuous diffusion further suggests that the discrete token representation is better aligned with the piecewise structure of manipulation actions.

