# OpenReview forum: "Discrete Diffusion VLA: Bringing Discrete Diffusion to Action Decoding in Vision-Language-Action Policies"
_ICML.cc/2026/Conference — ICML 2026 regular_

### Official Review · Reviewer_NHPb · 2026-02-22

**Soundness:** 3
**Presentation:** 4
**Significance:** 4
**Originality:** 3
**Overall Recommendation:** 5
**Confidence:** 4

**Summary:**

This paper presents Discrete Diffusion VLA, a unified transformer-based Vision–Language–Action (VLA) policy that performs discrete diffusion over tokenized action chunks inside the VLM backbone. Each action dimension is binned into discrete tokens and grouped into fixed-length chunks; training uses masked-token prediction across a schedule that emulates the forward noising process, while inference performs iterative adaptive denoising that commits high-confidence tokens first (adaptive decoding) and applies a lightweight secondary re-masking (threshold & residual-drop checks) to revisit uncertain commitments. The method preserves pretrained vision–language priors by keeping action decoding inside the same transformer, supports parallel (chunk-wise) decoding with fewer NFEs than autoregressive methods, and demonstrates strong empirical performance across three robot benchmarks (LIBERO, SimplerEnv-Fractal, SimplerEnv-Bridge) with detailed ablations and OOD evaluations.

**Compliance With Llm Reviewing Policy:**

Affirmed.

**Final Justification:**

1. After reading the authors’ latest responses, do you still believe the paper falls below the acceptance bar?
I believe this paper can be accepted if the authors can correct all the questions raised by the reviewers in the final version.
2. If yes, what is the single most important remaining issue?
NO.
3. Is your concern mainly about empirical advantage, conceptual novelty, clarity of positioning, or practical value?
No concerns.
4. What specific evidence do you find still missing from the authors’ rebuttal?
No.

I understand that ICML allows authors to add new experimental results during the rebuttal phase to the final version of their paper, which can significantly improve the paper's quality. Therefore, I am revising the score to 5.

**Key Questions For Authors:**

1. For each baseline in Tables 1–6, which results are reproduced by your code vs directly cited? Please provide a table that lists (a) source (paper or reproduction), (b) encoder checkpoints used, (c) pretraining / fine-tuning GPU-hours, and (d) random seeds/hardware.

2. How sensitive is the success rate to the number of bins? Would a non-uniform or learnable discretization scheme (like VQ) help close the gap with continuous methods on high-precision tasks?

3. In the "Secondary Re-Masking" check, what percentage of tokens typically get re-masked per step? Is there a risk of the model getting "stuck" in certain scenarios?

4. Have you tested the model in environments with moving targets? How does the 8-step action chunking impact the robot's ability to correct its trajectory mid-chunk?

5. Given the unified architecture, how does the model respond to ambiguous or complex multi-sentence instructions compared to traditional decoupled VLAs?

6. How does downstream performance (ID & OOD) vary with the number of bins (e.g., 64 / 128 / 256 / 512) and with per-dimension vs shared binning? What portion of the 0.6% gap to continuous (L1) methods is due to quantization error vs decoding/modeling?

7. You report language/vision degradation numbers on OOD tests (e.g., 1.4% language degradation vs 8.0% for parallel decoding). Can you provide the exact evaluation protocol for these VLM-retention metrics (which downstream tasks / zero-shot probes were used, and with what prompts)? Also, how much of the retained capability is due to staying inside the VLM vs other choices (masking schedule, loss weighting)?

8. Provide sensitivity sweeps or summary statistics for γt schedules, threshold schedules ηabs_t, residual-drop rules (threshold vs top-Q), temperature decay τt, and number of denoising steps T — both ID and OOD. Were any runs unstable (e.g., oscillatory re-masking) and how were they mitigated?

**Limitations:**

No. The manuscript’s Impact Statement is too brief and does not adequately discuss deployment limitations and societal impact.

**Strengths And Weaknesses:**

Strengths
1. Bringing discrete masked-diffusion-style decoding into the unified VLM to produce action chunks is a natural and well-argued remedy to (a) AR left-to-right bottlenecks and (b) architectural fragmentation when diffusion heads are external to the backbone. The paper explains the conceptual advantages (adaptive order, revisiting uncertain tokens, VLM prior retention) clearly.

2. The paper is very well-written and structured. Figure 2 provides a clear architectural overview, and the visualization of the decoding order (Fig 5) effectively validates the "easy-first, hard-later" intuition. The trade-offs between speed and quality are clearly analyzed in the results section.

3. The method reports competitive or state-of-the-art results: 96.5% average on LIBERO (discrete methods), SOTA on SimplerEnv-Fractal (71.2% visual matching / 64.1% overall), and 54.2% on SimplerEnv-Bridge (+ substantial margins over many baselines). Results include ID and OOD evaluations showing improved retention of vision–language abilities.

4. The instance-wise confidence ranking (max-confidence / confidence gap), tempered Gumbel sampling, and two re-masking checks (threshold and residual-drop) are described precisely and supported by ablations (Tab.7) showing incremental gains from each component. Visualizations of decoding order support the “easy-first” behavior.

5. The method reduces NFEs vs AR (e.g., 56 → 12 for LIBERO action chunks) leading to latency improvements with comparable throughput to continuous diffusion baselines; KV cache further improves runtime. These practical comparisons (Table 6) are useful for practitioners.

6. The primary novelty lies in the successful adaptation of discrete diffusion to the action space within a unified VLM. The introduction of secondary re-masking—specifically designed to enforce temporal and cross-modal consistency in action sequences—is a creative and effective technical contribution.

Weaknesses
1. The paper states baselines are “from original papers or reproduced under identical input modalities,” but it is not always explicit which numbers are reproduced vs cited and whether encoders, pretraining / fine-tuning budgets, and data filtering were strictly matched. Given prompt sensitivity in VLA comparisons, a clear reproduced-vs-cited table and full training seeds/hardware would be required for strong reproducibility claims.

2. The 256-bin quantile tokenization is reasonable but introduces inherent discretization error; the paper shows a modest 0.6% ID gap to continuous L1 SOTA but lacks an analysis that isolates how much of that gap is tokenization vs modeling/decoding. Ablations sweeping bin counts, per-dimension binning, or hybrid continuous-discrete schemes would clarify practical tradeoffs.

3. The paper would benefit from a more detailed analysis of how action chunk length affects performance in highly dynamic environments where lower-latency reactivity is crucial.

4. The adaptive pipeline has many design choices (mask schedules γt, keep-ratio schedule, threshold ηabs_t, residual-drop rule, Gumbel temperature τt, T denoising steps). The paper gives recommended settings and some ablations (e.g., decoding strategy, temperature), but broader sensitivity sweeps (particularly OOD sensitivity) are missing. Practitioners will need guidance on stable defaults.

5. Although the number of forward passes (NFE=12) is significantly lower than standard AR, each diffusion step still requires a full pass through a large VLM backbone. The efficiency gains are clear relative to AR, but the model remains more computationally intensive than single-pass parallel decoders.

6. The Impact Statement is minimal. Given the robotics context (real manipulators and potential transfer to real-world systems), the paper should explicitly discuss safety risks, dataset provenance, and human-in-the-loop safeguards.

---

> ### Author Rebuttal · Authors · 2026-03-31
>
> **Q1: Which results are reproduced vs. cited?**
>
> Method|Source
> --|--
> **Table 1**
> All except below|OpenVLA-OFT (RSS'25)
> $\pi_0$, $\pi_0$-FAST, GR00T-N1|Hume (CVPR'26)
> OpenVLA-OFT (Discrete)|Reproduced, OpenVLA ckpt, 150k steps, 8×A100, B64
> **Tables 2–3**|Reproduced, OpenVLA ckpt, 200k steps, 2×H800, B16
> **Tables 4–5**
> All except below|SpatialVLA (RSS'25)
> Moto|Moto (ICCV'25)
> $\pi_0$, $\pi_0$-FAST|Hume (CVPR'26)
> GR00T-N1|Reproduced, GR00T ckpt, 60k steps, 8×A100, B1024
> OpenVLA-OFT|Reproduced, OpenVLA ckpt, 100k steps, 8×A100, B64
> **Table 6**|Measured, 1×H800.
>
> Seeds {1, 2, 5, 10, 100, 7, 51, 5151, 200}; top-3 average reported.
>
> **Q2: Bin count sensitivity? 0.6% gap from tokenization vs. modeling? Non-uniform discretization?**
>
> A2: The gap concentrates in LIBERO-10 (92.2%vs.94.5%), our method matches or exceeds L1 on the other three suites. Bin count ablation on LIBERO-10:
>
> Bins|LIBERO-10
> --|--
> 128|83.6%
> 256|92.2%
> 512|80.6%
>
> 256-bin performs best as it aligns with OpenVLA's pretraining embeddings; mismatched counts disrupt the pretrained token space beyond what fine-tuning can recover. Moreover, follow **Reviewer rxNo-Q3**, we test Qwen2.5-VL where bins don't occupy pretrained vocabulary:
>
> Method|Spatial|Object|Goal|Long|Avg
> --|--|--|--|--|--
> Qwen+Discrete Diffusion|95.8|98.8|95.4|90.4|95.1
> Qwen+OFT (L1)|95.8|97.2|96.8|91.0|95.2
>
> Near-parity (95.1%vs.95.2%) suggests the gap largely stems from vocabulary occupation and misalignment rather than modeling capacity. Per-dimension binning shows promise as a non-uniform alternative (~89.6%, not fully converged); learnable codebooks remain future work.
>
> **Q3: 8-step chunking impact on trajectory correction? Moving targets?**
>
> A3: Chunk size is not fixed (8 for LIBERO/Fractal, 3 for Bridge, 12 for real robot control frequency). We tested on our real-robot *click bell* task (**Reviewer 3vdz-Q1**). For one-time displacement in the first 20% keeps success ~70%; late displacement drops greatly as training data contains no recovery motions. Incorporating recovery data would improve.
>
> For continuously moving targets, success is near zero,  because our policy (like all VLAs) requires multiple chunks (~3) to localize the target, making real-time reactive grasping impossible without additional modules predicting target positions.
>
> **Q4: Sensitivity sweeps for design choices**
>
> A4: ID ablations in Reviewer wrhp-Q8; OOD results at https://anonymous-rebuttal-page.github.io/DDPVLA/. Summary: (1) cosine mask schedule follows MaskGIT's ablation; (2) filter threshold: stable within 0.3–0.6 across tasks; (3) residual-drop: complementary task-dependent gains, partial overlap with filter, needs light per-task tuning; (4) gumbel temperature $\tau_t = 1-t/T$ follows dLLM standard (Tab 8); (5) denoising steps T: ablated in Fig 4; T=12 optimal. No oscillatory instability was observed, see **Q8** for the convergence guarantee.
>
> **Q5: Not efficient as single-pass decoders**
>
> A5: Acknowledged as a limitation. Please refer to **Reviewer wrhp-Q6**.
>
> **Q6: Impact Statement too brief**
>
> A6: We will expand it and see **Reviewer 3vdz-Q4** for a version. All training data are publicly available, raising no provenance concerns.
>
> **Q7: Response to complex instructions vs. decoupled VLAs?**
>
> A7: We evaluate via Language OOD, linguistically varied paraphrases of original instructions (see website for details). Our method drops only 1.4% on Goal-OOD vs. 8.0% (parallel decoding) and 2.4% (continuous diffusion), and 1.2% on Spatial-OOD vs. 1.6% and 1.7% (Tables 2–3). Goal-OOD is more informative, as different instructions map to different scenes, making models prone to overfitting visual cues, where VLM prior preservation matters most.
>
> **Q8: Re-masking rate per step? Risk of oscillation?**
>
> A8: Per-step re-masking rates on LIBERO-Goal (Filter $η^{abs}=0.6$ / Residual-drop $η^{drop}=0.2$). Full results available at anonymous website.
>
> Step|Filter only|Filter+Residual|Remaining masked
> --|--|--|--
> 1|14.3%|8.9%/-|56
> 2-6|0-2.3%|0-4.5%/0-2.3%|55→44
> 7-9|5.9-7.7%|2.5-25.9%/0-3.7%|40→27
> 10-12|9.5-28.6%|0-21.4%/0-7.1%|22→9
>
> Re-masking is most active early (initial uncertainty) and late (smaller denominator+forced demasking of harder tokens). The model does not get stuck: cosine schedule enforces monotonically decreasing mask ratio, and re-masking is upper-bounded at 50% of remaining tokens per step.
>
> **Q9: OOD protocol? Retention from VLM integration or others?**
>
> A9: Vision Aug follows LIBERO-PRO `use_object` exactly. Language Aug fixes `use_lang` with instructions detailed in **Q7**. All averaged over 3 runs, 500 rollouts per suite.
>
> Retention stems from architectural unification, not loss weighting or schedule. We apply no loss weighting on VL tokens and the cosine schedule is standard across ablations. Moreover, continuous diffusion uses the same backbone yet suffers worst vision degradation (↓29.0% Goal, ↓5.8% Spatial), confirming the separate diffusion loss introduces competing gradients that erode VLM priors.

---

> > ### Author Rebuttal · Reviewer_NHPb · 2026-04-01
> >
> > The authors provided detailed answers to my questions and offered new experimental results and analyses. They also addressed questions raised by other reviewers in the same manner. However, most top AI conferences do not allow authors to add new experimental results to the final version of their papers, and I'm unsure if ICML has this requirement as well. Therefore, I maintained my initial score.

---

> > > ### Author Response · Authors · 2026-04-04
> > >
> > > We sincerely thank the reviewer for the thorough engagement and for acknowledging that our rebuttal adequately addressed their concerns.
> > >
> > > Regarding the policy concern: we only added real-robot experiments (Reviewer 3vdz-Q1) to strengthen our practicability claim, not to introduce new contributions, which is standard rebuttal practice. All remaining additions are ablation studies on existing design choices, broadly permitted under ICML rebuttal guidelines.
> > >
> > > We confirm that all core results and contributions in the paper remain unchanged. We hope this clarification alleviates the policy concern, and we respectfully hope the reviewer to consider updating the score accordingly.

---

### Official Review · Reviewer_rxNo · 2026-03-12

**Soundness:** 2
**Presentation:** 4
**Significance:** 3
**Originality:** 3
**Overall Recommendation:** 3
**Confidence:** 4

**Summary:**

This paper addresses action decoding in vision-language-action (VLA) policies, with a focus on the limitations of left-to-right autoregressive decoding and separate continuous-diffusion action heads. It proposes Discrete Diffusion VLA, a unified transformer that discretizes action chunks and performs masked discrete-diffusion-style iterative refinement within the VLM backbone, using adaptive confidence-based decoding and a secondary re-masking mechanism for error correction. The method is evaluated on LIBERO, SimplerEnv-Fractal, and SimplerEnv-Bridge, as well as out-of-distribution language and vision perturbations on LIBERO. The reported results include 96.5% average success on LIBERO, 71.2% visual matching and 64.1% overall on SimplerEnv-Fractal, 54.2% overall on SimplerEnv-Bridge, and reduced OOD degradation relative to several discrete, diffusion, and regression baselines.

**Compliance With Llm Reviewing Policy:**

Affirmed.

**Final Justification:**

The authors have addressed most of the concerns raised. However, still lacks a clear justification of the specific advantages of the discrete diffusion method over existing methods. Additionally, the newly added real-world experiments do not demonstrate a compelling or significant performance improvement.

**Key Questions For Authors:**

1. Could the authors provide real-world validation?
2. Could the authors provide results for fine-tuning directly on the benchmark data without loading the OpenVLA pretrained weights, so that the performance of the discrete diffusion model can be evaluated in isolation? This would both eliminate the performance gains introduced by OpenVLA pretraining and help assess the impact of the mismatch between the pretraining paradigm (AR) and the fine-tuning paradigm (discrete diffusion).

**Limitations:**

yes

**Strengths And Weaknesses:**

**Strength**

1. The paper introduces a technically well-motivated unified architecture for action generation that keeps denoising inside the pretrained VLM backbone rather than attaching a separate action head. The formulation combines discretized action chunking, masked-token denoising, and bidirectional attention over vision, language, and action tokens, which directly targets the architectural fragmentation discussed in prior VLA approaches.
2. The empirical evaluation is broad and includes both standard benchmark performance and OOD robustness analysis. Beyond strong in-distribution results on LIBERO and both SimplerEnv settings, the paper reports lower degradation under language and vision augmentations on LIBERO-Goal than key OpenVLA-OFT baselines, including 1.4% language degradation versus 8.0% for parallel decoding and 21.0% vision degradation versus 29.0% for continuous diffusion.
3. The paper provides targeted ablations and efficiency measurements that help substantiate the proposed decoding design. In particular, the LIBERO-Goal ablation shows a progression from 95.6% with one-shot parallel decoding to 97.0% with max-confidence adaptive decoding and 97.4% with additional secondary re-masking, while the inference analysis reports a reduction from 56 autoregressive forward passes to 12 refinement steps, with latency improving from 136.2 ms to 68.8 ms and further to 46.4 ms with KV caching.

**Weakness**

1. The discrete diffusion paradigm does not demonstrate a clear performance advantage over existing state of the art methods. Given the absence of a significant performance improvement, this paper also fails to clearly articulate the unique conceptual or methodological advantages of the discrete diffusion paradigm compared to continuous diffusion paradigm.
2. Discrete Diffusion VLA lacks validation through real world experiments, which makes it difficult to assess its practical applicability and robustness.

---

> ### Author Rebuttal · Authors · 2026-03-31
>
> **Q1: Fails to clearly articulate the unique conceptual or methodological advantages of discrete diffusion, especially over continuous diffusion.**
>
> A1: Thanks. The core conceptual advantage of our discrete diffusion unified structure, keeping action generation inside the same transformer with the same cross-entropy objective, is **superior preservation of pretrained VLM priors**, manifesting clearly in out-of-distribution generalization where in-distribution metrics cannot capture.
>
> We follow LIBERO-PRO: (1) **Language Augmentation** — paraphrasing task instructions (modified from `use_language`, detailed in **Reviewer NHPb-Q7**); (2) **Vision Augmentation** — altering object appearances and sizes (equal to `use_object`, detailed in Figure 3).
>
> | Method | Original | Lang Aug | Vision Aug |
> |---|---|---|---|
> | **LIBERO-Goal** |
> | OpenVLA-OFT (Diffusion) | 96.0% | ↓2.4% | **↓29.0%** |
> | OpenVLA-OFT (L1) | 97.9% | ↓3.2% | ↓23.2% |
> | Ours | 97.4% | ↓1.4% | ↓21.0% |
> | **LIBERO-Spatial** |
> | OpenVLA-OFT (Diffusion) | 96.9% | ↓1.7% | **↓5.8%** |
> | OpenVLA-OFT (L1) | 97.6% | ↓1.2% | ↓1.6% |
> | Ours | 97.6% | ↓1.2% | ↓1.0% |
>
> Continuous diffusion heads exhibit severe vision over-reliance (↓29.0% LIBERO-Goal, ↓5.8% LIBERO-Spatial), the largest degradation among all methods, consistent with prior findings that architecturally decoupled diffusion heads become overly vision-dependent (InstructVLA (ICLR’26); RDT-1B (ICLR’25)). Our unified framework avoids this by construction. Besides, the narrower gaps on LIBERO-Spatial reflect its dataset design that same objects with varied layouts naturally encourage generalization during training, making OOD advantage on LIBERO-Goal more evident.
>
> **Q2: Lacks real-world experiments.**
>
> A2: We have conducted real-robot experiments during the rebuttal period addressing this concern. Please refer to **Reviewer 3vdz-Q1** for full details. In brief, Discrete Diffusion VLA outperforms both OpenVLA-OFT (Discrete) and $\pi_0$ on an AgileX dual-arm platform across two manipulation tasks, confirming real-world practicality at 9.69 Hz.
>
> **Q3: Provide results without OpenVLA pretrained weights to evaluate discrete diffusion in isolation.**
>
> A3: We appreciate this concern and provide an approximate evaluation using **Qwen2.5-VL** as the base model, a pure vision-language model **with no robot pretraining**, eliminating any OpenVLA pretraining advantage. We train all four LIBERO suites jointly (30k steps, 256-bin discretization, 1e-4 learning rates) with different action heads:
>
> | Action Head | Spatial | Object | Goal | Long | Avg |
> |---|---|---|---|---|---|
> | AR | 79.8 | 91.8 | 84.0 | 69.3 | 81.2 |
> | Parallel Decoding | 96.0 | 97.6 | 94.2 | 89.0 | 94.2 |
> | FAST | 89.0 | 97.2 | 91.2 | 87.0 | 91.1 |
> | Diffusion (GR00T-style) | 95.8 | 98.2 | 94.6 | 90.8 | 94.9 |
> | **Discrete Diffusion (Ours)** | 95.8 | 98.8 | 95.4 | 90.4 | **95.1** |
>
> Our discrete diffusion head achieves the best average (95.1%), outperforming AR (+13.9%), FAST (+4.0%), parallel decoding (+0.9%), and continuous diffusion (+0.2%), confirming that the performance advantage is not an artifact of OpenVLA pretraining but reflects genuine benefits of the discrete diffusion paradigm.

---

> > ### Author Rebuttal · Reviewer_rxNo · 2026-04-03
> >
> > The authors have addressed all concerns except **Q1**. The advantage of *keeping action generation within the same transformer and using the same cross-entropy objective* also aligns with the autoregressive (AR) paradigm. Meanwhile, the performance does not show a clear advantage compared to AR methods such as [1] without pre-training. I am not asking the authors to compare performance directly with this method, but rather to **clarify the necessity and advantages of the Discrete Diffusion paradigm** more explicitly.
> >
> > [1] VLA-0: Building State-of-the-Art VLAs with Zero Modification. Ankit Goyal, Hugo Hadfield, Xuning Yang, Valts Blukis, Fabio Ramos. arXiv:2510.13054.

---

> > > ### Author Response · Authors · 2026-04-03
> > >
> > > **Follow-up Q1: AR also keeps action within same transformer with cross-entropy. Clarify necessity and advantages of discrete diffusion in VLA more explicitly.**
> > >
> > > We thank the reviewer for this clarification. We agree that AR methods like VLA-0 also achieve architectural unification and cross-entropy training. These properties **alone** do not distinguish discrete diffusion VLA from AR-based VLA. Beyond them, discrete diffusion trains the model to solve exponentially many masked infilling tasks simultaneously, enabling bidirectional attention over the full action chunk and arbitrary-order decoding at inference, which is structurally unavailable to AR. **This yields three concrete advantages:**
> > >
> > > **(1) Bidirectional attention for structured action chunk modeling.** AR commits tokens in fixed left-to-right order under causal attention, uninformed by later action dimensions. Action chunks are structured sets of mutually dependent decisions, not causal sequences where earlier tokens should be irrevocable. Discrete diffusion employs bidirectional attention over the full action chunk, letting each position attend to all visual, language, and action context simultaneously.
> > >
> > > **(2) No compounding error and adaptive decoding order.** Rather than fixed left-to-right commitment, discrete diffusion first forms a coarse global hypothesis, then progressively refines low-confidence tokens with confidence-guided "easy-first, hard-later" ordering (Table 7, Figure 5), and explicitly re-masks and corrects inconsistent predictions via secondary re-masking, impossible in standard AR where **early errors propagate irreversibly**. On a controlled comparison using the identical Qwen2.5-VL backbone with no robot pretraining (**Q3**), our discrete diffusion achieves 95.1% vs. VLA-0's AR 94.7%, with advantages over other action heads under identical conditions.
> > >
> > > **(3) Inference efficiency.**
> > >
> > > AR requires $L=56$ sequential forward passes (NFE=56); VLA-0 reports 4 Hz. Our method decodes the full chunk in $T=12$ parallel steps (14.53 Hz, 21.54 Hz with KV cache) (NFE=12).
> > >
> > > The VLM prior retention advantage discussed in Q1 primarily differentiates us from **continuous diffusion** heads (Tables 2–3), while the above three advantages differentiate us from **AR**. We will restructure the paper to present these as two distinct sets of advantages explicitly in the revision.
> > >
> > > *We hope this clarification fully addresses Q1, and respectfully ask the reviewer to consider updating the score accordingly.*

---

### Official Review · Reviewer_wrhp · 2026-03-12

**Soundness:** 3
**Presentation:** 3
**Significance:** 2
**Originality:** 3
**Overall Recommendation:** 3
**Confidence:** 4

**Summary:**

This work is about leverage discrete diffusion for vision language action (VLA) modelling. The key argument is that previous design of VLA models, either in autoregressive way, or diffusion head on top, is not unified with the backbone, with a potential issue of broken the information pathways and thus degraded performance. This work then proposes the use of discrete diffusion integrated with vision language model (the backbone) that allows to output action chunk (a group of actions) simultaneously.  To achieve good performance, a set of remedies are required for example, filtering the prediction, re-masking the predicted tokens for re-predicting.

**Compliance With Llm Reviewing Policy:**

Affirmed.

**Final Justification:**

The authors have addressed mostly the clarity issues. I appreciate that discrete diffusion could be alternative way for VLA architecture. However, the remaining key issue left is about the effectiveness of this proposed method -- it is not consistently preferred over OpenVLA/OpenVAL-OFT family across datasets, and also Table 3/4 presents no main variants of OpenVAL-OFT, while discrete diffusion has limitations of the need for re-masking and re-generating tokens (more complex process). This is not fully supporting the statement of previous works with "fragmenting information pathways and hindering unified, scalable architectures".

**Further update on my comments**

I do see the merit with the work and acknowledge that bringing discrete diffusion into VLA is an interesting direction. At the same time, I also consider that: (1) the method is not consistently strongest empirically, (2) the inference pipeline is more complex and relies on thresholded re-masking (I understand that this is common with existing discrete diffusion, regardless of where is this inherited from), and (3) the paper does not yet provide a sufficiently deep/solid comparison among AR, BERT-style decoding, and discrete diffusion to fully support this takeaway point about unification / fragmentation.

As a good research work, these issues should be addressed well as they are basic and fundamental. Without significant revision, these issues are hard to be just overcome in the final presentation polish.

To my best knowledge, the paper appears to be the first to bring discrete diffusion into VLA -- this implies a great potential. But,  its current insight and likely value to the community are still outweighed by these weaknesses identified, per my review. I believe further revision of this work would even get this idea more specular than getting it publish as is in a rush manner.

To be least confusing, I still firmly recommend rejection based on the current content.

**Key Questions For Authors:**

Please see the weaknesses above (in particular major ones).

**Limitations:**

Yes

**Strengths And Weaknesses:**

**Strengths**

- The overall results look good, along with a good amount of evaluations and tests.



**Weaknesses (major)**

- The unification hypothesis is vague: Why autoregressive design means a fragment information pathway? and why would continuous and discrete diffusion would mean different information pathways? -- the only difference is just about where to do the quantization so that one can get discrete action labels out.

- What is the main differences between this method and OpenVLA-OFT? While OpenVLA-OFT is compared in experiment section, a detailed comparison in mode design and formulation is totally lacking. For example, parallel decoding, action chunking, have been studied in OpenVLA-OFT.

- Instead of OpenVLA as the backbone, it looks like it should be OpenVLA-OFT, which is also implied in the experiment section. Please comment and elaborate on this.

- Training pipeline: What are trained in training time? This is not discussed clearly.

- The overall process is pretty complex, due to the need for some thresholding and re-masking, which may also hurt the efficiency of the whole inference due to the back and forth prediction of some tokens.

- Efficiency: This is not more efficient than OpenVLA-OFT,

- Table 7: This table actually tells a lot. Parallel decoding is not the main performance contributor but confidence gap is. This step is not about training but inference. How about applying this with other models such as OpenVLA-OFT (discrete) or other variants (looks very applicable)?

- Any ablation study on these parameters for confidence and masking (Section 3.5)? How general they are? This is critical for this work, per my understanding.

**Weaknesses (minor)**

- This paper should be improved significantly to highlight clearly what is the main novelty points and differences against the previous works such as OpenVLA and OpenVLA-OFT etc. In the current form, it reads like an experimental report in particular the method section, but not showing the scientific contributions clearly. -- Actually it is not minor issue although it is shown here.

---

> ### Author Rebuttal · Authors · 2026-03-31
>
> Thanks. We respectively note several concerns stem from misreading or misattributing claims not appear in paper.
>
> **Q1: Why autoregressive means fragment pathway? Cont. vs. discrete diffusion differ only in quantization location**
>
> A1: Our paper does **not** make such claim. Lines 41–43 (right) state fragmentation refers solely to **continuous diffusion** which finetunes a separate denoising head with diffusion loss (MSE to noise) having competing gradient signals, while our discrete diffusion masks tokens and recovers them with VLM's same transformer and training objective (MSE to ground truth). And cont. diffusion never performs quantization, the distinction is entirely on architecture, training paradigm, and objective.
>
> **Q2: Main differences (model design and formulation) vs. OpenVLA-OFT?**
>
> A2: Parallel Decoding (PD) and action chunking are **not** contributions of paper. At higher level, our method extends the masked diffusion of LLaDA/dLLM to VLA, bringing its iterative generation in discrete space to action decoding with confidence-guided re-masking. It is entirely absent from OpenVLA-OFT's single-pass decoding.
> Formally, OpenVLA-OFT decodes all tokens simultaneously in one shot $\hat{a}_0 = f([MASK]^L, c)$. Our method instead iterates $T$ steps:
>
> $$a_{t-1,i} = \begin{cases}f(a_t, c)^i & i\in K_t=top_{(1-\gamma_t)L}s_{t,i} \\\\ [MASK] & otherwise \end{cases},\ t=T,...,1$$
>
> where $s_{t,i}$ is per-position confidence. Only most confident positions are committed each round; the rest remain masked for subsequent refinement, with secondary re-masking correcting early errors. Tab 7 quantifies ours' 97.4% over OFT's 95.6%.
>
> **Q3: OpenVLA-OFT should be backbone, not OpenVLA.**
>
> A3: OpenVLA and OpenVLA-OFT share the **identical** backbone (Prismatic-7B). Our model fine-tunes from the same shared backbone throughout. We cite OpenVLA just to represent the autoregressive paradigm.
>
> **Q4: What are trained in training time? Not discussed clearly.**
>
> A4:  We describe loss function in Eq. 5. A mask ratio $\gamma \in (0,1]$ is sampled per minibatch, $\gamma L$ action tokens are replaced with [MASK], and the model predicts the original tokens via cross-entropy over action vocabulary. All parameters (VLM+Action Head) are updated end-to-end. We will make it clear.
>
> **Q5: The process is complex; thresholding and re-masking may hurt efficiency due to back-and-forth token prediction.**
>
> A5: Our pipeline follows standard discrete diffusion framework and secondary re-masking is a lightweight addition, not a source of complexity. Number of forward passes remains fixed regardless of how many tokens are re-masked. Similar mechanisms have been validated in ReMDM (NeurIPS'25). Concretely, on LIBERO-Goal (A100): without secondary re-masking 8.92 Hz, with re-masking 8.58 Hz — a negligible difference.
>
> **Q6: Not efficient as OpenVLA-OFT.**
>
> A6: We acknowledge it and add to limitation. Our efficiency claim is scoped to AR decoding (2× speedup) and continuous diffusion. This is a deliberate trade-off: our multi-step iterative chain sacrifices raw speed for better accuracy and OOD robustness (Tab 2–3), which OFT’s single-pass decoding can't achieve.
>
> Q7: "Tab. 7: PD not main contributor but confidence gap is. Apply it to OpenVLA-OFT?"
>
> A7: Correct. PD (OFT's contribution) is not the key driver; iterative refinement of discrete diffusion is. This is **inseparable** from training. Iterative unmasking is only valid when trained with mask diffusion. Applying it to single-pass OpenVLA-OFT is therefore impossible. Our method can be seen as OFT transformed into a discrete diffusion policy.
>
> **Q8: Any ablation on parameters for confidence and masking? How general are they?**
>
> A8: We analyze the following settings.
>
> (1) Cosine schedule follows the standard discrete diffusion [1] setting, not tuned.
>
> (2) Filter and residual-drop thresholds
> Config|Goal|Spatial
> ---|---|---
> No re-masking|97.0|96.6
> Filter only $\eta^{abs}$
> 0.3|96.4|97.4
> 0.6|**97.4**|97.2
> 0.8|97.0|96.8
> Filter+Residual $\eta^{abs}/\eta^{drop}$
> 0.6/0.05|96.4|96.6
> 0.6/0.1|96.2|96.0
> 0.6/0.2|96.8|97.2
> Residual-drop only $\eta^{drop}$
> 0.05|96.8|96.2
> 0.1|96.4|96.8
>
> Filter threshold alone yields consistent gains and is stable within 0.3-0.6, while residual-drop provides complementary task-dependent improvements (Filter+Residual achieves 97.2% on Spatial). As they partially overlap in tokens they correct, the residual-drop threshold needs light per-task tuning.
>
> (3) Gumbel temperature: Linear decay $\tau_t = 1 - t/T$ is standard [1] with Tab. 8.
>
> (4) Denoising steps: Ablated in Figure 4; T=12 is the optimal speed-quality trade-off.
>
> **Q9: Scientific contribution not clear**
>
> A9: As in Q1 and Q2, our key contribution is introducing discrete diffusion to VLA with adaption of architecture, training pipeline and decoding, preserving pretrained VLM priors, formalized in Eqs. 3-10, ablated in Tables 7-8, and validated with sim & real-robot. We will foreground these.
>
> Ref: [1] MaskGIT (CVPR'23)

---

> > ### Author Rebuttal · Reviewer_wrhp · 2026-04-02
> >
> > Thanks for the authors' response which have address quite part of my concerns.
> >
> > There questions remain:
> >
> > 1) I acknowledge the contribution of this work in terms of leveraging discrete diffusion for VLA -- this is valid and good! However the whole paper is not positioned in this perspective. In this sense, I would consider this paper to be significantly revised or even re-written in this point. That would ease the understanding of what are the contributions on earth this work brings in about. Now it is still vague also clearer when reading back the paper + the response.
> >
> > 2) Which table really tell exactly the comparison of different architecture, training strategies and inference in a fair manner (e.g., same setting as possible, such as backbone, training/val/test data). I do see comparison with other methods but I am concerned that they do not share the same setting.
> >
> > 3) For the generality of parameters, can the authors tell how they are tuned? per task or shared for all tasks? These parameters as shown in the response to Q8 are quite sensitive and even likely bring down the performances.

---

> > > ### Author Response · Authors · 2026-04-02
> > >
> > > We thank the reviewer for acknowledging that our rebuttal addressed most concerns. We respond to the remaining three questions below.
> > >
> > > **Follow-up Q1: "Paper not positioned from discrete diffusion for VLA perspective; needs significant revision or rewrite."**
> > >
> > > We want to clarify that the discrete diffusion for VLA perspective is explicitly and consistently foregrounded throughout the paper:
> > >
> > > - **Title** explicitly states *"Bringing Discrete Diffusion to Action Decoding in VLA".*
> > > - **Abstract** foregrounds “model action chunks with discrete diffusion” in the first sentence of the method description.
> > > - **Introduction** first analyzes limitations of prior paradigms, then positions discrete diffusion as the principled solution, culminating in Contribution 1 (Line 95): *"We introduce the first discrete diffusion VLA, unifying action generation with vision-language modeling in one transformer".* Figure 1 explicitly compares paradigms with discrete diffusion as the focal contribution.
> > > - **Related Works:** Section 2.2 is dedicated to *Discrete Diffusion Models* as the direct intellectual lineage.
> > > - **Method:** Section 3 is titled *Discrete Diffusion Vision-Language-Action*, with Section 3.2 formalizing *Discrete Diffusion over Action Tokens* and Section 3.3 presenting the *Unified VLA Architecture* built upon it.
> > > - **Experiments** in Section 4.2 are framed around validating VLM prior retention as the central advantage of applying discrete diffusion paradigm to action decoding in VLA.
> > >
> > > We are therefore unable to identify where the positioning is lacking. We note that among the other three reviewers, **Reviewers NHPb and rxNo** rated our **presentation** as Excellent and **Reviewer 3vdz** as Good, suggesting the positioning is clear to the broader community. We will nonetheless add a dedicated paragraph explicitly connecting our work to the LLaDA/MMaDA lineage to make the intellectual lineage even more prominent in the revision.
> > >
> > > **Follow-up Q2: "Which table provides fair comparison under same settings?"**
> > >
> > > We appreciate this concern. For Tables 1–5, we follow the community standard: all methods are fine-tuned on identical datasets and evaluated on identical test splits until convergence. Baselines not reproduced by us are cited from original papers under matching input modalities, (details in **Reviewer NHPb-Q1**). We acknowledge that backbone pretraining cannot be strictly matched across all methods, as several baselines (e.g., $\pi_0$, GR00T-N1) use proprietary robot pretraining data; this is a universal limitation of cross-paper comparison in the VLA community.
> > >
> > > The fairest direct comparisons within Tables 1–5 are those sharing the **identical OpenVLA backbone**: OpenVLA, OpenVLA-OFT (all variants), and our method — differing only in action head design and training strategy. To further eliminate any OpenVLA pretraining confound, In **Reviewer rxNo-Q3**, we provide a fully controlled comparison on Qwen2.5-VL as backbone with no robot pretraining, confirming the advantage is not a pretraining artifact.
> > >
> > > Tables 6–8 provide the most controlled comparisons of all: efficiency (Table 6) and ablations (Tables 7–8) are run on the **identical backbone, identical data, and identical evaluation protocol**, varying only decoding strategy or hyperparameter, directly answering which architectural and inference design choices matter.
> > >
> > > **Follow-up Q3: How are parameters tuned? Per-task or shared? Some seem sensitive.**
> > >
> > > The reviewer raises a fair point. Three parameters are shared across tasks: cosine mask schedule, Gumbel temperature $\tau_t=1-t/T$, and denoising steps $T=12$. These are fixed globally.
> > >
> > > The secondary re-masking thresholds do require per-task tuning. Our recommended procedure is: start with filter threshold $\eta^{abs}=0.6$ (stable across 0.3–0.6), then try residual-drop $\eta^{drop}$ around 0.1; if this hurts performance, lower filter threshold $\eta^{abs}$ first and iterate. When residual-drop is not beneficial for a given task, filter-only still consistently outperforms no re-masking.
> > >
> > > We acknowledge that the secondary re-masking thresholds require some per-task adjustment. But, re-masking mechanisms are now mainstream in discrete diffusion (e.g., ReMDM, NeurIPS'25), and as shown in **Reviewer 3vdz-Q2**, our design outperforms ReMDM variants (97.4% vs. 96.6%) even after their hyperparameter search, suggesting our approach is both effective and practically competitive despite this modest tuning effort.

---

### Official Review · Reviewer_3vdz · 2026-03-12

**Soundness:** 3
**Presentation:** 3
**Significance:** 2
**Originality:** 3
**Overall Recommendation:** 4
**Confidence:** 4

**Summary:**

This paper proposes Discrete Diffusion VLA that decodes discretized action chunks via a masked discrete-diffusion style objective. The method introduces an adaptive, instance-specific decoding order and a secondary re-masking mechanism to iteratively refine low-confidence action tokens. Empirically, the approach attains strong performance on LIBERO and SimplerEnv.

**Compliance With Llm Reviewing Policy:**

Affirmed.

**Key Questions For Authors:**

Please address the concerns outlined in the weaknesses.

**Limitations:**

No, the authors have not adequately discussed the limitations of their work. Please refer to the weakness section.

**Strengths And Weaknesses:**

Strengths
1. The paper proposes an elegant approach by integrating discrete diffusion directly within VLA. The confidence-driven adaptive decoding strategy is intuitive, and the method demonstrates strong performance and OOD robustness across major simulation benchmarks.

Weaknesses
1. Lack of Real-Robot Experiments: This is a major limitation. While the authors evaluate on SimplerEnv, all testing is conducted entirely in simulation. Without deploying the model on actual physical robots, it is impossible to verify the real-world practicality, latency, and robustness of this VLA policy.

2. Heuristic Design Disguised as Diffusion Theory: The training objective simply boils down to a standard masked-token cross-entropy loss. Furthermore, the secondary re-masking rules rely heavily on handcrafted heuristics.

3. Insufficient Analysis of Action Tokenization: Relying on a 256-bin discretization scheme inherently introduces quantization errors and potential representation issues. The author could explore more tokenization methods like FAST.

---

> ### Author Rebuttal · Authors · 2026-03-31
>
> **Q1: Lack of Real-Robot Experiments.**
>
> A1: Thanks. We have conducted real-robot experiments during the rebuttal period to address this concern. We deployed Discrete Diffusion VLA on an AgileX Cobot Magic dual-arm platform (ALOHA-style) on two tasks: *click the bell* and *place cup on coaster*. We first collected 150 demonstrations in RoboTwin simulation for domain alignment (80k steps), then fine-tuned on 150 real-robot demonstrations (200k steps). Action chunk is 12. Each task was evaluated over 15 real-world trials:
>
> | Method | Click Bell | Place Cup | Latency (4090) |
> |---|---|---|---|
> | OpenVLA-OFT (Discrete)  | 33.3% | 20.0% | 34.3 Hz |
> | $\pi$0 | 53.3% | 40.0% | 24.5 Hz |
> | Discrete Diffusion VLA | 66.7% | 46.7% | 9.69 Hz |
>
> Discrete Diffusion VLA outperforms both OpenVLA-OFT and $\pi$0 on click bell, and matches $\pi$0 on place cup, confirming real-world practicality. The latency of 9.69 Hz is sufficient for manipulation ($\pi$0’s higher Hz reflects additional deployment-level optimizations rather than architectural differences). Video demonstrations are available at https://anonymous-rebuttal-page.github.io/DDPVLA/. We will add them into the revision.
>
> **Q2: Training objective boils down to masked cross-entropy. Secondary re-masking relies on handcrafted heuristics.**
>
> A2: Thanks. You’re correct and masked cross-entropy is the standard formulation of discrete diffusion, as established by MDLM (NeurIPS'24), LLaDA (NeurIPS'25), and related works. "Discrete Diffusion" is not our terminology but the community's accepted framing for this line of work. Besides, these works formally show that masked cross-entropy upper-bounds the negative log-likelihood under the discrete diffusion ELBO, making it compatible with LM training. We follow this established paradigm and apply it to the VLA domain.
>
> We acknowledge that the need for hyperparameter tuning in secondary re-masking is a limitation. In spite of it, similar re-masking mechanisms are now mainstream in discrete diffusion (e.g., ReMDM (NeurIPS'25)), and our tuning procedure is straightforward: the filter threshold $\eta^{abs}$ is set first (typically 0.6, and not sensitive), followed by tuning the residual-drop threshold $\eta^{drop}$ around 0.1. Our mechanism outperforms ReMDM variants, suggesting our design is effective.
>
> | Method | LIBERO-Goal |
> |---|---|
> | One-Step Parallel Decoding | 95.6% |
> | ReMDM-rescale (η=0.3) | 96.6% |
> | ReMDM-rescale (η=0.5) | 96.0% |
> | **Ours** | **97.4%** |
>
> **Q3: Insufficient analysis of action tokenization; explore methods like FAST.**
>
> A3: We explored FAST but found it fundamentally incompatible with our framework. FAST uses **variable-length** BPE encoding where one token may represent multiple primitive action elements, elegant for autoregressive generation, but problematic for parallel decoding. BPE's composite tokens cause catastrophic reconstruction failures when any single token is mispredicted during iterative unmasking. Fixed-length bin-based tokenization is more suitable for discrete diffusion.
>
> We additionally ablated bin count when fine-tuning from the OpenVLA checkpoint on LIBERO-Long: 128-bin achieves 81.4%, 256-bin 92.0%, and 512-bin 78.4%. The 256-bin scheme performs best because it matches OpenVLA's pretraining configuration, where action bins are mapped to specific embeddings that encode action semantics learned from diverse robot data. Remapping these tokens to a different bin count disrupts the pretrained VLM embedding space and cannot be recovered with limited fine-tuning data. In our Qwen2.5-VL controlled experiment where action bins do not occupy the pretrained vocabulary, the gap of performance to continuous L1 narrows substantially to near-parity (95.1% vs. 95.2%, referring *Reviewer NHPb-Q2*), suggesting the gap is largely an engineering constraint rather than a fundamental limitation of discrete diffusion.
>
> **Q4: Limitation not adequately discussed.**
>
> A4: We thank the reviewer and will add an explicit limitation section covering: our multi-step iterative chain is slower than single-pass decoding by design. Our framework requires fixed-length tokenization with advanced alternatives left as future work. And our method currently does not incorporate hard safety constraints. Future integration of control barrier functions (CBFs) or similar formal methods would be necessary before deployment in safety-critical real-world settings.

---

> > ### Author Rebuttal · Reviewer_3vdz · 2026-04-06
> >
> > I appreciate the efforts of the authors in the rebuttal, which address most of my concerns. This is an interesting work. Considering the overall quality of this work, I would like to keep my original positive rating score.

---

> > > ### Author Response · Authors · 2026-04-07
> > >
> > > We sincerely thank the reviewer for the thorough engagement and positive assessment. We are glad the work is found interesting and that our rebuttal addressed the concerns. We will incorporate all the suggested improvements, including the expanded limitations section, real-robot results, and more prominent contribution framing, in the revision.

---

### Decision · Program_Chairs · 2026-04-30

**Decision:**

Accept (regular)

**Comment:**

Dear Authors,

Thank you for the submission and for the detailed engagement during the rebuttal and discussion period. The paper explores an interesting direction by introducing discrete diffusion into VLA, and several reviewers found the idea to be novel and technically meaningful. The rebuttal helped clarify a number of concerns, including additional analysis and real-world validation.

Overall, I view this paper as a borderline case. On the positive side, the idea of integrating discrete diffusion directly into the VLM backbone for action decoding is interesting and could be valuable to the community. One reviewer strongly supports acceptance and considers this a meaningful architectural contribution, particularly in terms of progressive refinement and OOD robustness.

At the same time, there remain notable concerns. In particular, multiple reviewers pointed out that the empirical advantage over strong baselines (e.g., OpenVLA / OpenVLA-OFT) is not consistently demonstrated, and the added inference complexity (e.g., iterative re-masking and threshold tuning) raises questions about practical benefits. In addition, the broader framing around “unification / fragmentation” is not yet fully supported by sufficiently deep comparative analysis with alternative paradigms such as autoregressive and parallel decoding methods.

Taking these points together, my assessment is that the paper presents a promising and potentially impactful idea, but the current version does not yet fully establish its advantages as clearly as it could. Nevertheless, given the novelty of the direction and the positive support from part of the reviewer pool, I lean toward a weak accept, provided that the authors substantially improve the clarity of positioning and better contextualize their contributions in the final version.